# The prevalence, prevention, and treatment of cardiovascular diseases in Twelve African Countries (2014–2019): An analysis of the World Health Organisation STEPwise approach to chronic disease risk factor surveillance

**Wingston Felix Ng'ambi**[1]*, **Janne Estill**[1], **Fatma Aziza Merzouki**[1], **Cosamas Zyambo**[2], **Jonathan Chiwanda Banda**[3], **David Beran**[4], **Olivia Keiser**[1]

1 Institute of Global Health, University of Geneva, Geneva, Switzerland, 2 Department of Community and Family Medicine, University of Zambia, Lusaka, Zambia, 3 Division of Non-Communicable Diseases and Mental Health, Ministry of Health, Lilongwe, Malawi, 4 Division of Tropical and Humanitarian Medicine, University of Geneva, Geneva, Switzerland

* wingston.ngambi@gmail.com

## Abstract

### Introduction

Cardiovascular diseases (CVDs) are responsible for nearly a third of deaths globally. We conducted this study to understand the prevalence of history of CVDs, their prevention and treatment in twelve African countries using the World Health Organization STEPwise Approach to Surveillance (WHO STEPS) data.

### Methods

We used secondary STEPS data extracted from 12 African countries between 2014 and 2019. CVD was defined as a self-reported history of heart attack, angina, or stroke. Weighted percentages, counts, weighted odds ratios (OR), and the corresponding 95% confidence intervals (95%CI) were computed using the R software. We fitted logistic regression models to select the predictor variables from a regression model for CVD prevalence, CVD prevention and CVD treatment binary endpoints.

### Results

Amongst 60,294 individuals, the prevalence of CVD was 5%. The CVD prevalence was higher in older individuals, females, individuals with hypertension, smokers, people with high salt intake, and in certain countries. Eleven percent of the 23,630 individuals at high risk of CVD (≥40 years) but without a history of the disease received CVD prevention treatment. Amongst the 2,895 persons with CVDs, 22% received treatment and counselling for CVD: 34% (n = 215) receiving aspirin, 32%

**Data availability statement:** The data used in this study may be accessed at https://extranet.who.int/ncdsmicrodata/index.php/catalog/629.

**Funding:** The author(s) received no specific funding for this work.

**Competing interests:** We declare no competing interest.

(n = 202) counselling for CVD risk factors, 11% (n = 66) statins, and 24% (n = 148) both statins and aspirin. The uptake of CVD treatment varied by hypertension status, sex, age and country.

## Conclusion

The prevalence of CVD was relatively low and CVD treatment uptake was sub-optimal. Concerted efforts must be made to accelerate the diagnosis and expand treatment for CVDs in Africa if to curtail untimely deaths attributable to CVDs.

## Introduction

Despite the relatively low prevalence of cardiovascular diseases (CVDs) in Africa compared to developed countries, there is evidence of a steady increase, which poses a significant public health concern. One of the aims of the United Nations Sustainable Development Goals is to reduce premature mortality from non-communicable diseases (NCDs) by a third by the year 2030 [1]. The CVDs, like coronary heart disease and stroke, are the most common NCDs globally responsible for an estimated 9.6 million male deaths and 8.9 million female deaths worldwide in 2019, of which 6.1 million of were among people aged 30–70 years [2]. Of these deaths related to CVDs, more than 75% occurred in low- and middle-income countries [3]. Heart attacks and strokes accounted for 85% of these fatalities [4]. Globally between 1990 and 2020 the age standardized DALYs for CVDs equal those of the communicable, maternal, neonatal and nutritional (CMNN) diseases combined.

While infectious diseases have historically been the primary focus of public health efforts in the region, CVDs are becoming a significant and increasing problem [5]. The prevalence of CVDs in Africa has been steadily increasing over the past few decades [5]. Despite the growing prevalence of CVDs in Africa, there is a lack of multi-country analyses assessing the full care continuum; from diagnosis to treatment; in this region. According to the Global Burden of Disease (GBD) project, prevalence of CVDs and mortality differ substantially between high-income and low-income regions. Particularly, Africa is seeing rising rates of diabetes, obesity, and hypertension; all of which are significant causes of CVD; but these patterns are not well-represented in international research initiatives. The region is understudied in comparison to more resourced locations because of issues with infrastructure, data availability, and healthcare systems [6]. Furthermore, while global trends in prevalence of CVDs and their associated risk factors have been extensively studied; Africa remains relatively understudied despite the rising burden of CVDs [6]. A dearth of comprehensive and localised studies that are suited to the region's particular socioeconomic and healthcare contexts is further highlighted by recent reports from GBD collaborations that emphasise the need to shed light on the CVD epidemiology in Africa. Through region-specific analysis, these initiatives are starting to address modifiable risk factors; nevertheless, there are still gaps in assessing the effectiveness and implementation of interventions [7]. Efforts to address this growing health

concern, like improved healthcare access, awareness campaigns, and research to develop region-specific prevention and treatment strategies, are critical [8,9]. For example, the World Health Organization developed the package of essential noncommunicable disease interventions (PEN) and related strategies (PEN-PLUS) for resource limited settings covering also the care of severe NCDs in these settings [2,10].

The prevalence of CVDs in Africa is influenced by a complex interplay of various determinants such as hypertension, diet, physical activity, tobacco use, alcohol consumption, age, gender, genetic factors, socio-economic deprivation (including poverty and limited access to healthcare), environmental factors (pollution and limited access to clean and safe water, and exposure to toxins), lack of awareness about CVD risk factors and prevention, and limited availability and quality of healthcare services and facilities in the region. These factors can significantly influence the diagnosis, treatment, and prevention of CVD [11,12]. In addition, the prevalence of CVDs in Africa is influenced by multiple determinants that vary across the regions, making it important to consider both spatial and temporal aspects when addressing the burden of CVDs in the region. The study acknowledges the complex interplay of determinants influencing the prevalence of CVDs in Africa, including socio-economic, environmental, and health system factors. This approach highlights the importance of considering regional variability, which is often overlooked in global analyses. By concentrating on Africa, the study fills a crucial gap in the literature, particularly in the context of limited healthcare resources and overlapping burdens of infectious diseases like HIV/AIDS and tuberculosis. Understanding the CVD determinants in Africa is crucial for developing effective strategies to reduce the prevalence of CVDs in Africa. For example, cessation of tobacco use, reduction of salt in the diet, eating more fruit and vegetables, regular physical activity, and avoiding harmful use of alcohol have been shown to be protective of CVD [13]. Furthermore, identifying the persons at highest risk of CVDs and ensuring they receive appropriate treatment can prevent premature deaths [14]. While the WHO STEPwise approach has provided a framework for monitoring NCDs, its data remain underutilized, especially in Africa [15]. This study addresses a critical gap in global CVD research by providing the first multi-country analysis of CVD prevalence, prevention, and treatment in Africa using WHO STEPS data. Its findings are not only relevant for Africa but also provide a template for utilizing similar data in other regions, thus contributing to the global effort to reduce NCD-related premature mortality.

## Methods

### Study design and setting

This is a secondary analysis of WHO STEPS data from countries with CVD data collected between 2014 and 2019 in Africa (see Box 1). This analysis includes data from 12 countries in Africa: Algeria, Benin, Botswana, Eswatini, Ethiopia, Kenya, Malawi, Morocco, São Tomé and Príncipe, Sudan, Uganda, and Zambia. The WHO STEPS assess risk factors for chronic non-communicable diseases and uses a multi-stage cluster sampling of households. One individual within the age range of the survey (18–69 years) was selected per household [16]. For example, Malawi's STEPS survey in 2017 and Zambia's STEPS survey in 2017 followed this methodology to capture critical health data for analysis [17,18]. The WHO STEPS use simple, standardized methods for collecting, analysing and disseminating data on key NCD risk factors in the countries. The survey covers key behavioural risk factors: tobacco use, alcohol use, physical inactivity, and unhealthy diet; as well as key biological risk factors: overweight and obesity, raised blood pressure, raised blood glucose, and abnormal blood lipids [16]. The survey is conducted using a stepwise procedure that begins with a questionnaire to collect data on risk factors (STEPS 1), progresses to basic physical examinations (STEPS 2), and concludes with the more intricate collection of blood samples for biochemical analysis (STEPS 3) [16]. The 2014–2019 period was selected because data on CVD were unavailable before this timeframe.

### WHO package of essential noncommunicable intervention implementation

Across Africa, countries exhibit considerable heterogeneity in the timing and extent of adopting national NCD strategies and implementing the WHO Package of Essential Noncommunicable (PEN) disease interventions. The WHO PEN

**Box 1. WHO STEPWise Surveys (STEPS 1, 2 & 3) with cardiovascular disease data in African countries: 2014–2019.**

| Country | Africa region | Survey year | Records included |
|---|---|---|---|
| Algeria | Northern | 2016 | 6,955 |
| Benin | Western | 2015 | 5,115 |
| Botswana | Southern | 2014 | 3,888 |
| Eswatini | Southern | 2014 | 3,026 |
| Ethiopia | Eastern | 2015 | 9,241 |
| Kenya | Eastern | 2015 | 4,477 |
| Malawi | Eastern | 2017 | 4,186 |
| Morocco | Northern | 2017 | 4,991 |
| Sao Tome and Principe | Central | 2019 | 2,418 |
| Sudan | Northern | 2016 | 7,722 |
| Uganda | Eastern | 2014 | 3,973 |
| Zambia | Eastern | 2017 | 4,302 |
| Total | | | 60,294 |

approach was first introduced globally in 2010 and WHO/AFRO has supported member states in PEN roll-out since at least 2008 [19]. In North Africa, Algeria formalized its national NCD strategic planning most recently (with an updated UNSDCF including NCDs from 2023) and Morocco has integrated PEN elements into its primary care cancer prevention efforts, with routine risk factor surveillance reported since 2018. In West Africa, Benin's earlier NCD strategic plan (2014–2018) provides the backbone for current pilot PEN activities, and Sao Tome and Principe incorporated PEN into its NCD action planning upon joining WHO/AFRO's regional frameworks in 2023. In East Africa, Ethiopia's NCD strategic planning dates back to at least 2014–2016, followed by updated strategies into the early 2020s; PEN activities have been scaled in selected regions with quarterly monitoring. Kenya's national NCD strategy (2015–2020) underpins systematic PEN roll-out across counties, and Uganda's policy (published 2020) supports PEN training for frontline staff. Southern African countries show similar evolutionary progress: Botswana reports PEN guideline integration into primary care with service improvements; Eswatini's strategic NCD plan dates from the 2012–2020 era; and Zambia has embedded PEN within its NCD framework, recently launching a PEN-Plus National Operational Plan in 2025 to expand care and training beyond primary care. Sudan adopted an NCD strategy as early as 2010–2015, though political instability has constrained consistent PEN implementation. In Malawi, PEN integration into the National Health Strategic Plan has guided initial district hospital training since the early 2020s, with ongoing expansion to peripheral centres.

**Variables and data management**

The most recent WHO STEPS data on CVD risk factors were extracted from each country. The data were managed in R (see 1_**Create_dataset_for_analysis.R; Functions_rmph.R**).

   **Outcome variables.**  The primary outcome variable was whether an individual responded "yes" or "no" to the question "Have you ever had a heart attack, angina (chest discomfort caused by heart disease), or stroke (cerebrovascular accident or incident)?", which we considered as a proxy for having CVD. We excluded 141,833 patients with missing CVD information. Prevention of CVD or uptake of treatment for CVD was assessed with the questions "Are you currently taking aspirin regularly to prevent or treat heart disease?" (yes/no), "Are you currently taking statins (Lovostatin/Simvastatin/

 

Atorvastatin or any other statin) regularly to prevent or treat heart disease?" (yes/no) or "During the past three years, has a doctor or other health worker advised you to do any of the following: quit using tobacco or not start, reduce alcohol consumption or not start, reduce salt in your diet, reduce refined sugar in your diet, or eat at least five servings of fruits and/or vegetables each day?" (yes/no).

**Predictor variables.** The predictor variables comprised the following known dichotomous risk and protective factors defined as follows: harmful alcohol use (defined as daily drinking and having at least 4 (for males) or 3 (for females) drinking occasions in the last 30 days) derived using the Alcohol Use Disorder Identification Test-C (AUDIT-C) [20]; low consumption of vegetables (i.e., "In a typical week, on how many days do you eat vegetables?" with less than five times a week being categorized as low consumption of vegetables); low consumption of fruits (i.e., "In a typical week, on how many days do you eat fruit?" with less than five times a week being categorized to have low fruit consumption); physical inactivity (less than 150 minutes of moderate-intensity physical activity per week) [21]; high salt intake (i.e., "How often do you add salt or a salty sauce such as soy sauce to your food right before you eat it or as you are eating it?" with those adding salt to food regularly or often eating processed food with high salt quantities are considered to have high salt intake); history of raised blood pressure (systolic blood pressure ≥160 and/or diastolic blood pressure ≥ 100 mmHg or currently on medication); and history of raised blood glucose (capillary whole blood value at least 6.1 mmol/L) [22]. We also included smoking history (currently smoking, previously smoking and never smoked); body mass index (<18.5, 18.5–24.0, 25.0–29.0, 30.0 + kg/m$^2$; excluding pregnant women), sex of the respondent (male, female), age (15–29, 30–39, 40–49, 50–59, 60 +; in completed calendar years), highest education level (none, primary, secondary, tertiary), occupation (government employee, non-government employee, self-employed, retired, or not working (including, e.g., students, unemployed)), type of residence (rural, urban), country, and marital status (never married, married, previously married).

The distribution of missingness in the analysed data is shown in Box 2. Overall, 26% of the individuals had missing data. The variable with the greatest missing data was diabetes with 8% missing data. Imputation was performed to handle the missing data because it allows for the inclusion of all available data in the analysis, ensuring more accurate and representative results. Different probability distributions were specified depending on the type of variable. For binary variables, such as presence or absence of a condition, we used a binomial distribution, which models the probability of a "success" or "failure" outcome. For categorical variables with more than two groups, such as education level or occupation, we applied a multinomial distribution that accounts for multiple outcome categories and ensures that the probabilities across categories sum to one. For continuous variables, including age, body mass index, and blood pressure, a Gaussian distribution was used to generate values based on the mean and variance of the observed data. These simulation models were implemented using random assignment based on the observed distribution of each variable, ensuring that imputed values reflected the empirical patterns in the data. This strategy allowed us to approximate the underlying data-generating mechanism more realistically, reduce bias from listwise deletion, and retain the full analytic sample.

We opted for simulation-based imputation instead of standard MICE approaches for several reasons. First, the data exhibited complex patterns across multiple countries, including high inter-variable correlations and heterogeneity in variable distributions, which led to convergence issues and unstable chains when using MICE. Second, some variables had non-standard distributions or rare categories that are not easily handled by default MICE models. Third, simulation-based imputation allows direct specification of the appropriate distribution for each variable type (binomial, multinomial, Gaussian), ensuring that imputed values reflect the empirical distribution of the observed data. Missing categorical data were imputed using the base R function sample(), drawing from the observed distribution of non-missing values, while continuous variables such as age were simulated using rnorm() based on the observed mean and standard deviation, with results rounded to the nearest integer. Replacement values were therefore drawn from overall empirical distributions, and we acknowledge that this approach does not capture country-specific heterogeneity in risk factor distributions. To support reproducibility and ensure plausibility, we set a random seed prior to simulation

**Box 2. Proportion of missing data across variables used to assess cardiovascular disease prevalence in twelve African countries, 2014–2019.**

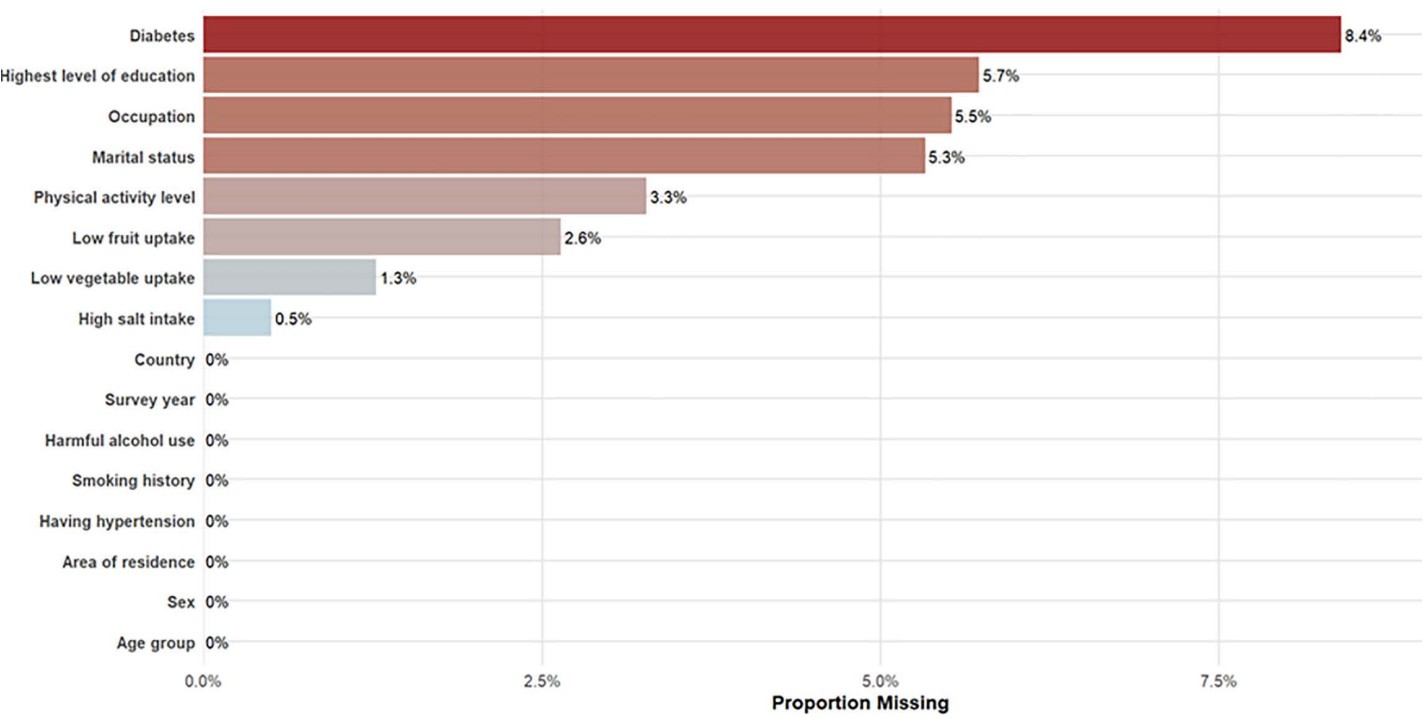

and compared the distribution of imputed values with complete-case data (see **sample simulation code below or 2_who_steps_ncd_data_cleaning_code_2024.R).**

Finally, this approach avoids the iterative dependency structure of MICE, which can amplify biases in the presence of autocorrelation, while preserving sample size and statistical power. Furthermore, imputation is a common practice in epidemiological research to maintain data integrity and reduce bias due to incomplete datasets [23]. In order to ascertain the uptake of CVD prevention treatment, all the individuals aged at least 40 years were considered to be at risk of CVD and this formed the denominator for this analysis [24,25,26,27,28]. Furthermore, the WHO STEPS data analysis guide also considers the CVD naïve individuals aged at least 40 years as being at CVD risk [29,30].

**Data analysisa**

We set up the survey design before fitting any models and used the same weight variable, wstep1, throughout the analysis (see 3_**CVD_Analysis_SSA_2024_FINAL.Rmd**). For the CVD prevalence and prevention datasets, we defined a full complex design using primary sampling unit (psu) as the cluster, stratum as the stratification variable, and wstep1 as the sampling weight, with nesting enabled. For the treatment dataset, only wstep1 was available, so we applied a simple one stage design. All quasibinomial models for CVD prevalence were run on these survey design objects, ensuring that the sampling structure was properly accounted for. We also adjusted for inter country variation by including country fixed effects, rather than treating countries as random clusters. This allowed us to capture differences across countries in a clear and consistent way.

Our analysis is structured around the CVD care pathway; spanning diagnosis, treatment initiation, and counselling/ adherence support; which provides a systematic framework to identify gaps in care and is particularly relevant in the African context, where resource constraints and variations in health system capacity can lead to substantial drop-offs at each stage of the cascade (see Box 3). Weighted percentages, counts, weighted odds ratios (OR), and the corresponding 95% confidence intervals (95%CI) were computed using the R software (version 4.3.2). The weighting variable from WHO STEPS was based on demographic characteristics of the general population from which a sample was taken (wstep1) [31]. We conducted a set of three weighted univariable and multivariable logistic regression analyses, with quasibinomial distributions [32], of the effects of the predictor variables on: CVD prevalence in the whole dataset, CVD prevention amongst the persons at risk of CVD, and CVD treatment (and counselling) among those with CVD (see 3_**CVD_Analysis_SSA_2024_FINAL.Rmd**).

The quasibinomial distribution, an extension of the standard binomial distribution, is particularly suitable for survey data with binary outcomes, as it accounts for overdispersion (see 3_**CVD_Analysis_SSA_2024_FINAL.Rmd**). For each survey-weighted multivariable quasibinomial logistic regression model, survey weights were incorporated directly into the

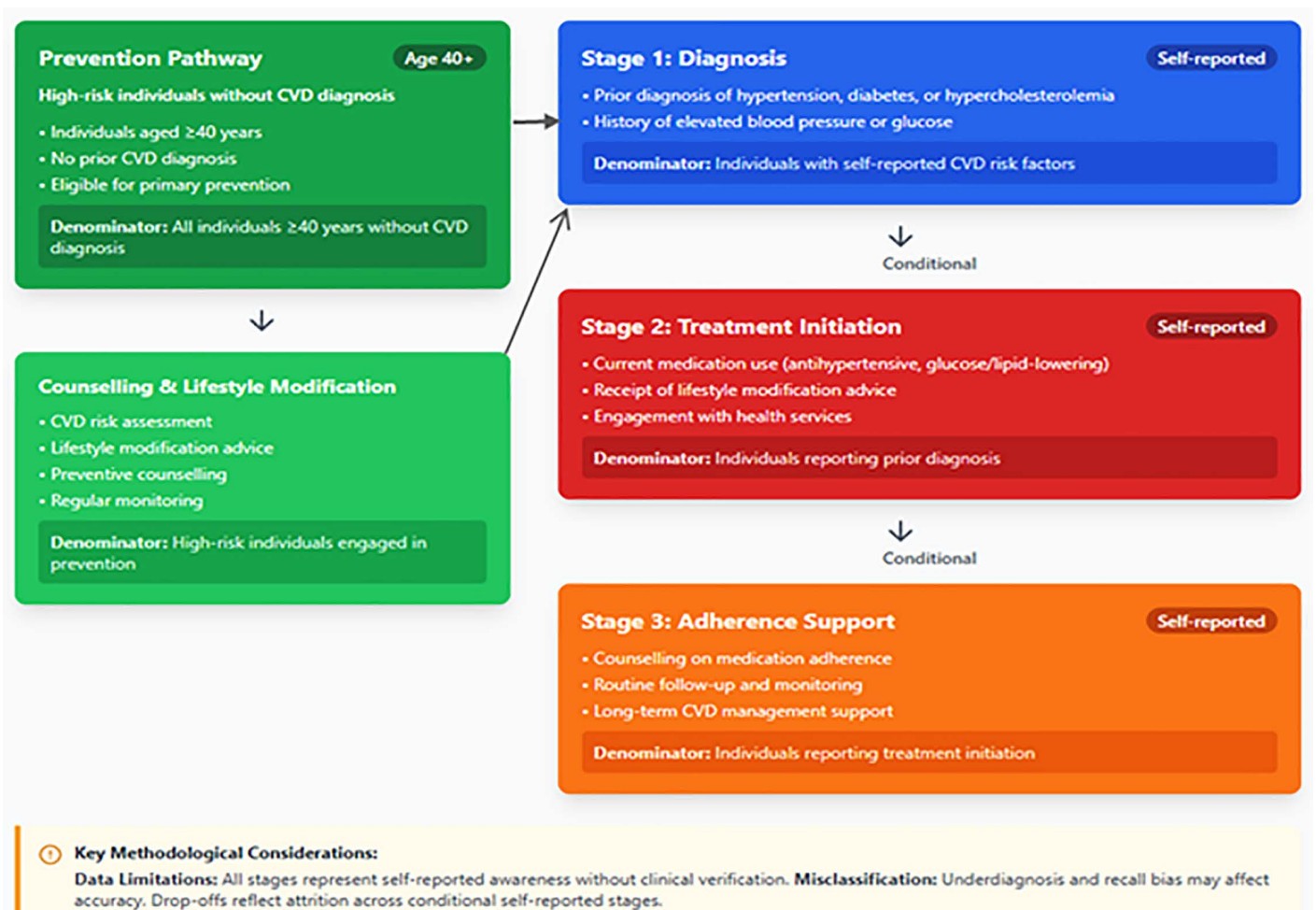

**Box 3. The Cardiovascular disease (CVD) care pathway from WHO STEPWise Surveys conducted across African countries (2014–2019). CVD = cardiovascular diseases.**

model fitting to ensure that parameter estimates reflected the complex sampling design. Weighted standard errors were calculated to account for unequal selection probabilities, thereby producing valid confidence intervals and p-values. We first fitted a full model including all candidate predictors and then applied backward variable selection using the stepAIC function from the MASS R package, retaining predictors that optimized model fit while respecting the survey weights [33]. The stepAIC function was used for stepwise selection, optimizing model fit by retaining predictors that minimized the AIC. Notably, the clinically important variables; age, sex, and hypertension status; were naturally retained in the final model based on the AIC criterion, so no variables needed to be forced into the model. We calculated the P-values for the bivariate logistic regression by comparing the model with the variable of interest and the empty model. Weighted quasibinomial logistic regression models with country fixed effects were used to account for country-level heterogeneity, as mixed-effects models were not preferred due to heterogeneity in survey design and instability with weighted data. Similarly, we calculated the P-values from adjusted models by comparing the final model with the model without the variable of interest. Wherever applicable the level of statistical significance was set at P<0.05.

Sensitivity analyses (see 3_**4_CVD sensitivity analysis 2025.R**) were conducted to assess the robustness of the imputation assumptions by comparing results from the simulation-based imputation to a complete case analysis as shown in Box 4. This approach evaluated whether the main findings were influenced by the distributional assumptions used for imputing missing values. Based on the results there was perfect agreement of the coefficients from the imputed and complete case analyses coefficients (see Box 4).

### Ethical consideration

Individual written consent was obtained from all participants before data collection the during the original WHO STEPS surveys. We requested access for secondary use of the data from the World Health Organization, the funder of the WHO STEPS surveys. The WHO STEPS survey datasets were downloaded from https://extranet.who.int/ncdsmicrodata/index.php/catalog/629 [22]. This dataset is anonymized with no identifiable information on survey respondents.

**Box 4. Sensitivity analysis of determinant coefficients for cardiovascular disease prevalence, comparing imputation and complete case analysis in twelve African countries, 2014–2019.**

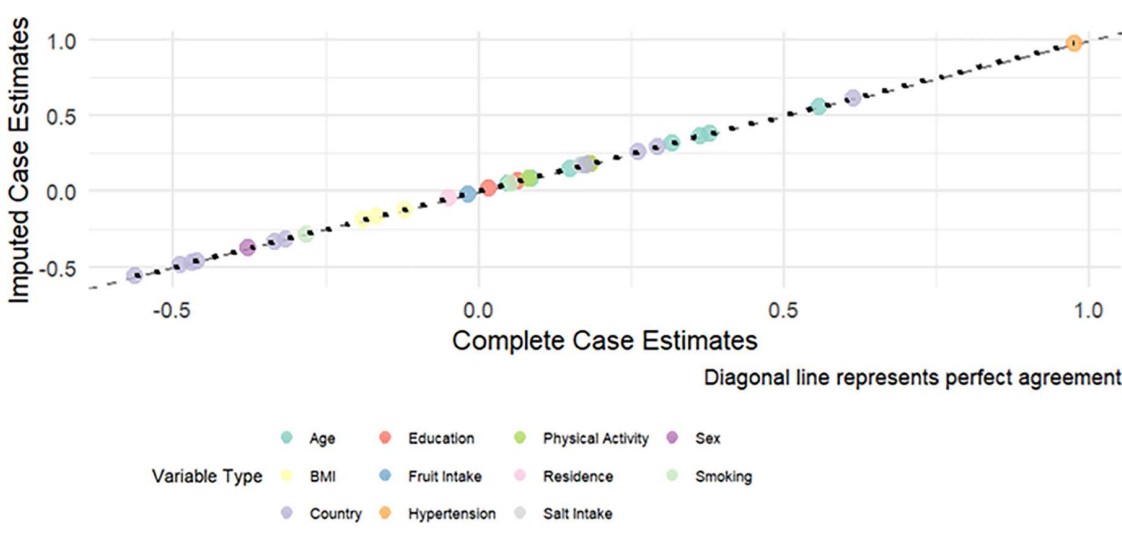

## Results

### Individuals included and excluded from the study

The distribution of individuals included in the study is shown in Fig 1. Of the 202,127 individuals included in the database, 60,294 (29.8%) were interviewed from 2014 onwards and included information on CVD.

### Characteristics of the study population

The characteristics of individuals included in this study are shown in Table 1. Of the 60,294 individuals, 44% were aged 15–29 years while 6% were aged 60 years or above. The proportions of males and females were similar. More individuals were from rural areas (60%) than urban areas (40%). About a third of the individuals had no education and almost half of the individuals were self-employed (see Table 1). The majority of individuals were currently married. The countries with highest numbers of respondents were Ethiopia (23%), Algeria (21%), and Sudan (16%).

Of the 60,294 individuals, 9% had hypertension, 6% had high cholesterol and 3% had diabetes. Furthermore, 19,333 (33%) reported high salt uptake, 45,207 (81%) low fruit uptake, and 28,905 (54%) low vegetable uptake (Table 1). The prevalence of low level of physical activity was reported as 16% while the prevalence of reported high level of physical activity was 65%. A total of 1,803 (3%) of the 60,294 individuals reported using alcohol in a harmful way. Furthermore, 5,047 (10%) and 3348 (7%) of the 60,294 individuals reported being current and previous smokers, respectively. The prevalence of reported current smoking was 10% while 6% of the individuals had previously smoked or used tobacco. The prevalence of measured underweight was 10% and the prevalence of overweight (i.e., BMI of at least $30\,kg/m^2$) was 14%. The highest number of individuals were interviewed in 2015 or 2016 and the least in 2019.

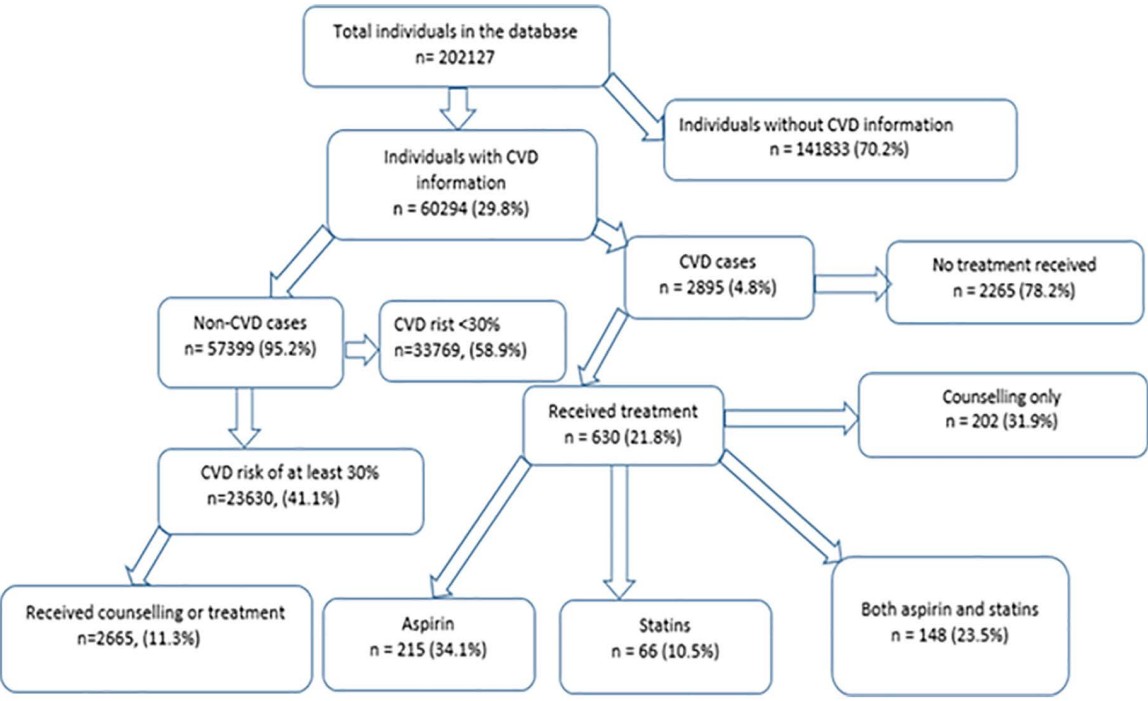

**Fig 1. Flow diagram of individuals included and excluded in the analysis: 2014–2019.**

**Table 1. Characteristics of the study population, the CVD cases, and treatment uptake in twelve African countries between 2014 and 2019.** The percentages for the study population are column percentages; for CVD cases and patients who received CVD treatment, the percentages are row percentages with the denominator being the whole study population within the respective category for CVD cases, and CVD cases within the respective category for patients who received CVD treatment.

| Characteristics | Study population[a] | | CVD cases[b] | | Received CVD treatment[c] | |
|---|---|---|---|---|---|---|
| | n | % | n | % | n | % |
| Total | 60294 | 100.0 | 2895 | 4.6 | 630 | 18.5 |
| **Age group** | | | | | | |
| *15-29* | 19351 | 44.1 | 748 | 3.6 | 71 | 9.2 |
| *30-39* | 15853 | 23.0 | 687 | 4.6 | 81 | 7.9 |
| *40-49* | 11563 | 16.6 | 553 | 5.1 | 109 | 19.4 |
| *50-59* | 7929 | 10.3 | 484 | 6.2 | 173 | 36.5 |
| *60+* | 5598 | 6.0 | 423 | 8.4 | 196 | 45.4 |
| **Sex** | | | | | | |
| *Female* | 37010 | 48.8 | 1961 | 5.6 | 430 | 18.2 |
| *Male* | 23284 | 51.2 | 934 | 3.7 | 200 | 19.0 |
| **Area of residence** | | | | | | |
| *Rural* | 33718 | 59.7 | 1590 | 4.3 | 258 | 13.1 |
| *Urban* | 26576 | 40.3 | 1305 | 5.1 | 372 | 25.2 |
| **Highest level of education** | | | | | | |
| *None* | 20174 | 29.0 | 947 | 4.4 | 229 | 20.3 |
| *Primary* | 19647 | 35.0 | 1057 | 5.5 | 220 | 17.1 |
| *Secondary* | 14915 | 25.7 | 655 | 4.0 | 116 | 16.7 |
| *Tertiary* | 5558 | 10.2 | 236 | 3.8 | 65 | 24.4 |
| **Occupation** | | | | | | |
| *Government employee* | 4413 | 7.8 | 203 | 3.9 | 59 | 23.3 |
| *Non-Govt employee* | 4622 | 8.7 | 210 | 3.7 | 45 | 21.6 |
| *Non-paid/Retired* | 25400 | 36.2 | 1180 | 5.2 | 289 | 20.6 |
| *Self-employed* | 25859 | 47.3 | 1302 | 4.5 | 237 | 15.5 |
| **Marital status** | | | | | | |
| *Currently married* | 39949 | 68.5 | 2060 | 4.9 | 474 | 19.9 |
| *Formerly married* | 4019 | 5.4 | 243 | 6.8 | 49 | 12.8 |
| *Never married* | 16326 | 26.1 | 592 | 3.4 | 107 | 15.6 |
| **Having diabetes** | | | | | | |
| *No* | 58800 | 97.4 | 2814 | 4.6 | 608 | 18.4 |
| *Yes* | 1494 | 2.6 | 81 | 4.1 | 22 | 23.6 |
| **Having high cholesterol level** | | | | | | |
| *Yes* | 5926 | 6.2 | 279 | 5.0 | 79 | 18.6 |
| *No* | 54368 | 93.8 | 2616 | 4.6 | 551 | 18.5 |
| **Having hypertension** | | | | | | |
| *No* | 52772 | 91.5 | 2100 | 3.9 | 218 | 9.1 |
| *Yes* | 7522 | 8.5 | 795 | 11.8 | 412 | 52.6 |
| **High salt intake** | | | | | | |
| *No* | 40961 | 66.8 | 2005 | 4.7 | 467 | 21.0 |
| *Yes* | 19333 | 33.2 | 890 | 4.5 | 163 | 13.1 |
| **Low fruit uptake** | | | | | | |

*(Continued)*

| Characteristics | Study population[a] | | CVD cases[b] | | Received CVD treatment[c] | |
|---|---|---|---|---|---|---|
| | n | % | n | % | n | % |
| *No* | 14037 | 18.7 | 719 | 4.9 | 188 | 24.0 |
| *Yes* | 46257 | 81.3 | 2176 | 4.5 | 442 | 17.1 |
| **Low vegetable uptake** | | | | | | |
| *No* | 31130 | 46.3 | 1511 | 4.9 | 384 | 24.5 |
| *Yes* | 29164 | 53.7 | 1384 | 4.4 | 246 | 12.7 |
| **Physical activity level** | | | | | | |
| *Low Level* | 12325 | 16.4 | 562 | 4.6 | 199 | 34.5 |
| *Moderate level* | 12725 | 19.7 | 565 | 4.4 | 153 | 23.9 |
| *High level* | 35244 | 63.9 | 1768 | 4.7 | 278 | 12.9 |
| **Smoking history** | | | | | | |
| *Current* | 5047 | 10.2 | 258 | 5.2 | 45 | 16.4 |
| *Never* | 51899 | 83.2 | 2425 | 4.4 | 518 | 17.6 |
| *Previous* | 3348 | 6.7 | 212 | 5.8 | 67 | 29.6 |
| **Harmful alcohol use** | | | | | | |
| *No* | 58491 | 97.1 | 2775 | 4.6 | 615 | 18.7 |
| *Yes* | 1803 | 2.9 | 120 | 5.4 | 15 | 14.0 |
| **BMI** | | | | | | |
| *<18.5* | 7899 | 15.5 | 331 | 3.5 | 50 | 12.9 |
| *18.5-24* | 29966 | 53.0 | 1426 | 4.7 | 199 | 10.8 |
| *25-29* | 12417 | 17.5 | 617 | 5.2 | 180 | 27.4 |
| *30+* | 10012 | 14.0 | 521 | 4.9 | 201 | 38.8 |
| **Survey year** | | | | | | |
| *2014* | 10887 | 10.8 | 712 | 8.7 | 109 | 10.6 |
| *2015* | 18833 | 39.2 | 926 | 4.3 | 116 | 8.0 |
| *2016* | 14677 | 37.3 | 532 | 3.7 | 232 | 38.6 |
| *2017* | 13479 | 12.5 | 635 | 4.8 | 147 | 14.7 |
| *2019* | 2418 | 0.2 | 90 | 3.4 | 26 | 27.9 |

[a]**is the denominator for b when calculating the prevalence of CVD cases**

[b]**is the denominator for c when calculating uptake of CVD treatment amongst the CVD cases**

## Care cascade for the cardiovascular disease

Across the 12 study countries, a substantial gap was observed along the CVD care cascade in Fig 2. While the total eligible population ranged widely, only a small proportion had a documented diagnosis of CVD, and an even smaller fraction received treatment. For example, in Algeria, out of 6,955 individuals, only 415 (6.0%) had a CVD diagnosis and 178 (2.6%) were on treatment. Similarly, in Malawi, 312 of 4,186 participants (7.5%) were diagnosed, but just 64 (1.5%) received treatment. Diagnosis rates were particularly low in Kenya (0.4%) and Eswatini (4.3%), while Morocco showed the highest diagnosis proportion (3.3%). Treatment coverage among those diagnosed also varied widely, with Algeria achieving 42.9%, Morocco 36.2%, and Zambia only 15.0%. Overall, the cascade illustrates that fewer than one in ten individuals are diagnosed, and less than half of those diagnosed receive treatment, underscoring major gaps in CVD detection and management across the region.

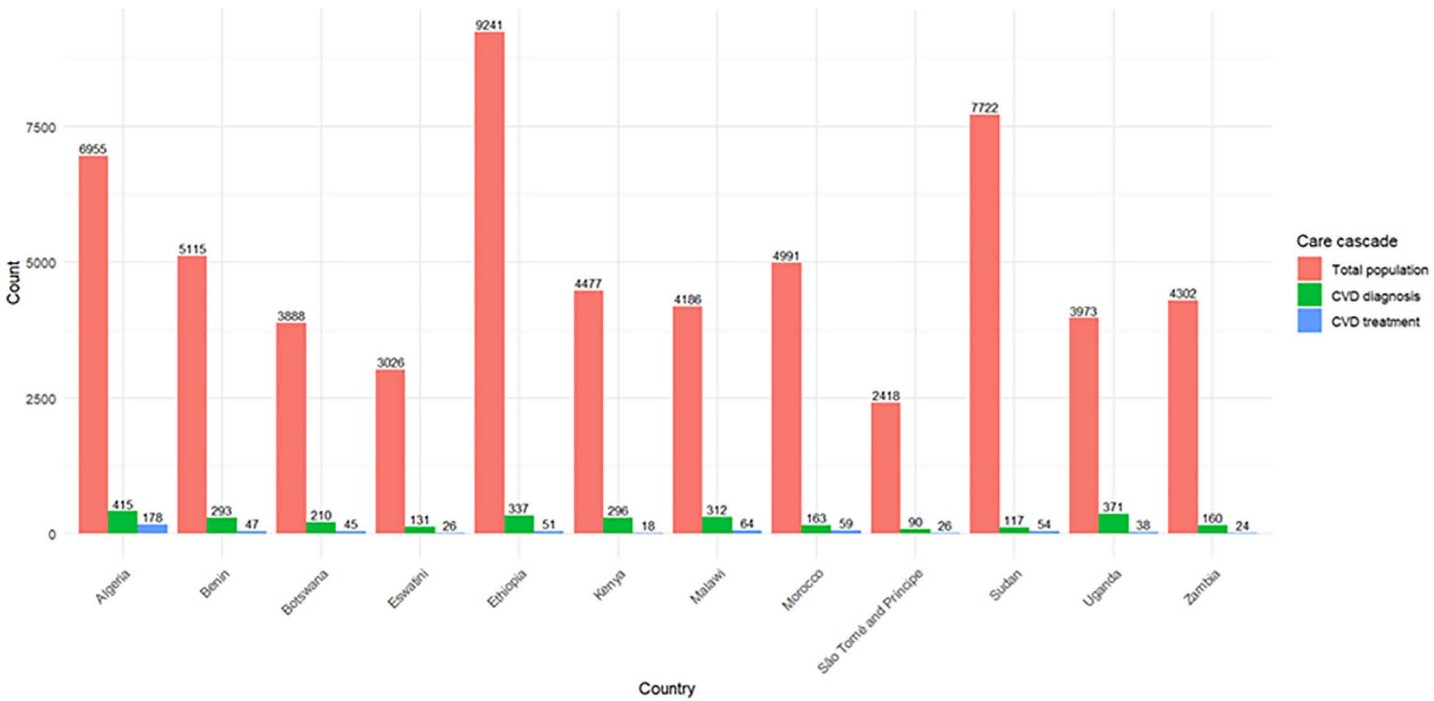

**Fig 2. Cardiovascular disease care cascade in twelve African countries, 2014–2019.**

### Prevalence of CVD and factors associated with CVD in twelve countries of Africa

The prevalence of CVD was 5% (2895/60,294) and increased with age (Table 1). The prevalence of CVD was higher in females than males, in urban than rural areas, and in individuals with hypertension, using alcohol harmfully, or being obese or overweight. Uganda (9%), Malawi (7%), Botswana (6%), Algeria (6%) and Kenya (6%) had the highest, and Sudan (1%), Zambia (3%), and Morocco (3%) the lowest prevalence of CVD (Table 1).

Older age, male sex, certain countries of residence, having hypertension, high salt uptake, and current or previous smoking were significantly associated with higher risk of CVD. The adjusted estimates show that there was an increasing trend in CVD with age (Fig 3). After adjusting for sex, area of residence (urban vs rural), highest education level, marital status, country of residence, having hypertension, high salt intake, physical activity level, and smoking status, the individuals aged 50−59 or at least 60 years were more likely to have CVD (aOR: 1.39, 95%CI: 1.17–1.66 and aOR: 1.67, 95%CI: 1.36–2.07, respectively) than those aged 15−29 years. Males were less likely to have CVD than females (aOR: 0.65, 95%CI: 0.54–0.72, P<0.001). Country-specific differences highlight the role of regional factors, with Sudan showing the lowest risk and Uganda the highest (Table 2). Individuals with hypertension were almost three times more likely to have CVD compared to those without hypertension (aOR: 2.68, 95%CI: 2.32–3.10, P<0.001). Similarly, individuals with high salt uptake were 23% more likely to have CVD than those without high salt uptake (aOR: 1.23, 95%CI: 1.02–1.49, P=0.03). Those who never smoked tobacco were less likely to have CVD compared to those currently smoking (aOR: 0.69, 95%CI: 0.55–0.86, P<0.0001).

### Uptake of CVD prevention amongst the individuals aged at least 40 years in twelve countries of Africa

The characteristics of the 23,630 individuals who were at risk of CVD and interviewed on the uptake of CVD prevention treatment and counselling are shown in Table 2. Of these 23,630 individuals, the majority (51%) were aged between 40

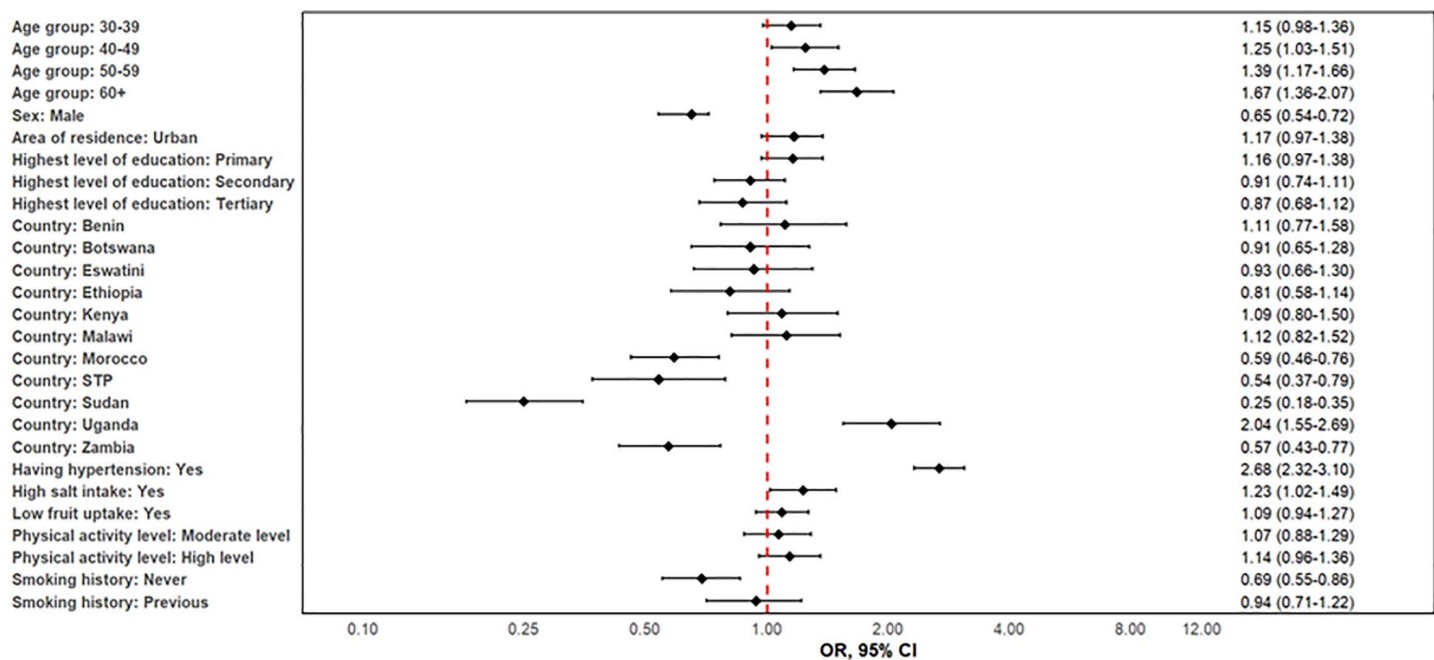

**Fig 3. Determinants of cardiovascular disease prevalence in twelve African countries, 2014–2019.**

and 49 years, 53% were males, 56% were from rural areas, 37% had no education, 27% were from Algeria while less than 0.1% were from Morocco, 3% had diabetes, 9% had high cholesterol level, 16% had hypertension, 29% had high salt uptake, 81% had low fruit uptake, 49% had low vegetable uptake, 20% had low physical activity, 12% were current tobacco smokers, and 4% were harmful alcohol users.

Only 11% received any form of prevention or counselling, highlighting a major prevention gap. The distribution of the uptake of CVD prevention by the characteristics of the respondents is shown in Fig 4. People aged at least 60 years, those with BMI of at least 30, those who had previously smoked tobacco, those with low physical activity and those with tertiary education had the highest uptake of CVD prevention. Similarly, higher uptake of CVD prevention was observed amongst the females, urban residents, those with diabetes, those with hypertension, those with low salt uptake, and those who were not harmful alcohol users.

The factors associated with the uptake of CVD prevention are shown in Fig 4. Individuals who were aged at least 50 years were more likely to get CVD prevention compared to those who were aged between 40 and 49 years. Males were less likely to get CVD prevention compared to females (aOR: 0.73, 95%CI: 0.58–0.932, P<0.0001). Urban residents are more likely to receive prevention (aOR: 1.65, 95%CI: 1.33–2.04, P<0.0001) than their rural counterparts. Educational attainment influences uptake, with individuals having a tertiary education showing the highest uptake (Fig 4). Previous smokers had higher (aOR: 1.17, 95% CI: 0.83–1.64) and non-smokers lower uptake (aOR: 0.88, 95% CI: 0.64–1.20) compared with current smokers. Individuals with hypertension were 17 times more likely to get CVD prevention compared to those without hypertension (aOR: 16.65, 95%CI:13.93–19.91, P<0.0001). There were also country variations in uptake of treatment for preventing CVDs (see Fig 4). These findings suggest that targeted interventions addressing these modifiable factors could improve CVD prevention uptake in Sub-Saharan Africa.

**Table 2. Characteristics of persons who were at risk of cardiovascular diseases, the uptake of prevention of CVD and the factors associated with the uptake of prevention services for CVD in Africa between 2014 and 2019.**

| Characteristics | Total | | Received prevention | |
|---|---|---|---|---|
| | n | % | n | % |
| Total | 23630 | 100 | 2665 | 11.3 |
| **Age group** | | | | |
| *40-49* | 11010 | 51 | 711 | 5.8 |
| *50-59* | 7445 | 31.2 | 997 | 11.6 |
| *60+* | 5175 | 17.8 | 957 | 16.6 |
| **Sex** | | | | |
| *Female* | 13763 | 46.6 | 1893 | 12.7 |
| *Male* | 9867 | 53.4 | 772 | 6.8 |
| **Area of residence** | | | | |
| *Rural* | 12560 | 55.9 | 878 | 5.1 |
| *Urban* | 11070 | 44.1 | 1787 | 15.2 |
| **Highest level of education** | | | | |
| *None* | 10123 | 36.5 | 996 | 6.8 |
| *Primary* | 7526 | 34.8 | 825 | 9.5 |
| *Secondary* | 4281 | 20.6 | 554 | 11.7 |
| *Tertiary* | 1700 | 8.1 | 290 | 16.7 |
| **Occupation** | | | | |
| *Government employee* | 1974 | 10.2 | 289 | 13 |
| *Non-Govt employee* | 1748 | 8.6 | 160 | 8.8 |
| *Non-paid/Retired* | 9905 | 36.3 | 1450 | 12.9 |
| *Self-employed* | 10003 | 44.9 | 766 | 6.2 |
| **Marital status** | | | | |
| *Currently married* | 18714 | 85.9 | 2032 | 9.6 |
| *Formerly married* | 2296 | 7.8 | 291 | 9.8 |
| *Never married* | 2620 | 6.3 | 342 | 8.9 |
| **Country** | | | | |
| *Algeria* | 3375 | 27.1 | 725 | 20.1 |
| *Benin* | 1894 | 1.7 | 66 | 3.8 |
| *Botswana* | 1408 | 0.5 | 336 | 21.7 |
| *Eswatini* | 1189 | 0.2 | 191 | 15.6 |
| *Ethiopia* | 3105 | 20.5 | 133 | 2.7 |
| *Kenya* | 1626 | 13 | 80 | 4.9 |
| *Malawi* | 1518 | 5.8 | 77 | 4.5 |
| *Morocco* | 2680 | 0 | 426 | 14.2 |
| *STP* | 897 | 0.2 | 115 | 13.8 |
| *Sudan* | 3219 | 16.6 | 415 | 11.8 |
| *Uganda* | 1184 | 8.9 | 41 | 2.9 |
| *Zambia* | 1535 | 5.5 | 60 | 4 |
| **Having diabetes** | | | | |
| *No* | 23012 | 97 | 2561 | 9.4 |
| *Yes* | 618 | 3 | 104 | 14.6 |
| **Having high cholesterol level** | | | | |

*(Continued)*

Table 2. (Continued)

| Characteristics | Total | | Received prevention | |
|---|---|---|---|---|
| | n | % | n | % |
| *Yes* | 2899 | 8.7 | 433 | 11.7 |
| *No* | 20731 | 91.3 | 2232 | 9.3 |
| **Having hypertension** | | | | |
| *No* | 19015 | 84.3 | 618 | 3.4 |
| *Yes* | 4615 | 15.7 | 2047 | 42.6 |
| **High salt intake** | | | | |
| *No* | 16856 | 70.7 | 2000 | 11.4 |
| *Yes* | 6774 | 29.3 | 665 | 5 |
| **Low fruit uptake** | | | | |
| *No* | 5693 | 19.4 | 843 | 14.4 |
| *Yes* | 17937 | 80.6 | 1822 | 8.4 |
| **Low vegetable uptake** | | | | |
| *No* | 13099 | 50.9 | 1776 | 12.7 |
| *Yes* | 10531 | 49.1 | 889 | 6.3 |
| **Physical activity level** | | | | |
| *Low Level* | 5588 | 20.2 | 928 | 16.2 |
| *Moderate level* | 5013 | 20.3 | 732 | 13.3 |
| *High level* | 13029 | 59.5 | 1005 | 6 |
| **Smoking history** | | | | |
| *Current* | 2342 | 11.7 | 158 | 6.4 |
| *Never* | 19338 | 77.2 | 2252 | 9.6 |
| *Previous* | 1950 | 11.1 | 255 | 12.2 |
| **Harmful alcohol use** | | | | |
| *No* | 22800 | 96.2 | 2633 | 9.8 |
| *Yes* | 830 | 3.8 | 32 | 2.5 |
| **BMI** | | | | |
| *<18.5* | 2990 | 14.9 | 195 | 5.7 |
| *18.5-24* | 10199 | 46.1 | 637 | 5.1 |
| *25-29* | 5612 | 21.4 | 808 | 13.9 |
| *30+* | 4829 | 17.6 | 1025 | 19.2 |
| **Survey year** | | | | |
| *2014* | 3781 | 9.6 | 568 | 4.1 |
| *2015* | 6625 | 35.2 | 279 | 3.6 |
| *2016* | 6594 | 43.6 | 1140 | 16.9 |
| *2017* | 5733 | 11.4 | 563 | 4.3 |
| *2019* | 897 | 0.2 | 115 | 13.8 |

STP= Sao Tome and Principe; BMI= Body mass index

## Uptake of treatment for CVDs in twelve countries of Africa

The treatment and counselling received by the respondents with CVD is shown in Fig 1. A total of 630 (22%) of the 2895 individuals with CVD received treatment and counselling. Of these 630 patients, 32% received counselling, 34% received aspirin (but no statins), 11% received statins (but no aspirin), and 24% received both statins and aspirin. The distribution

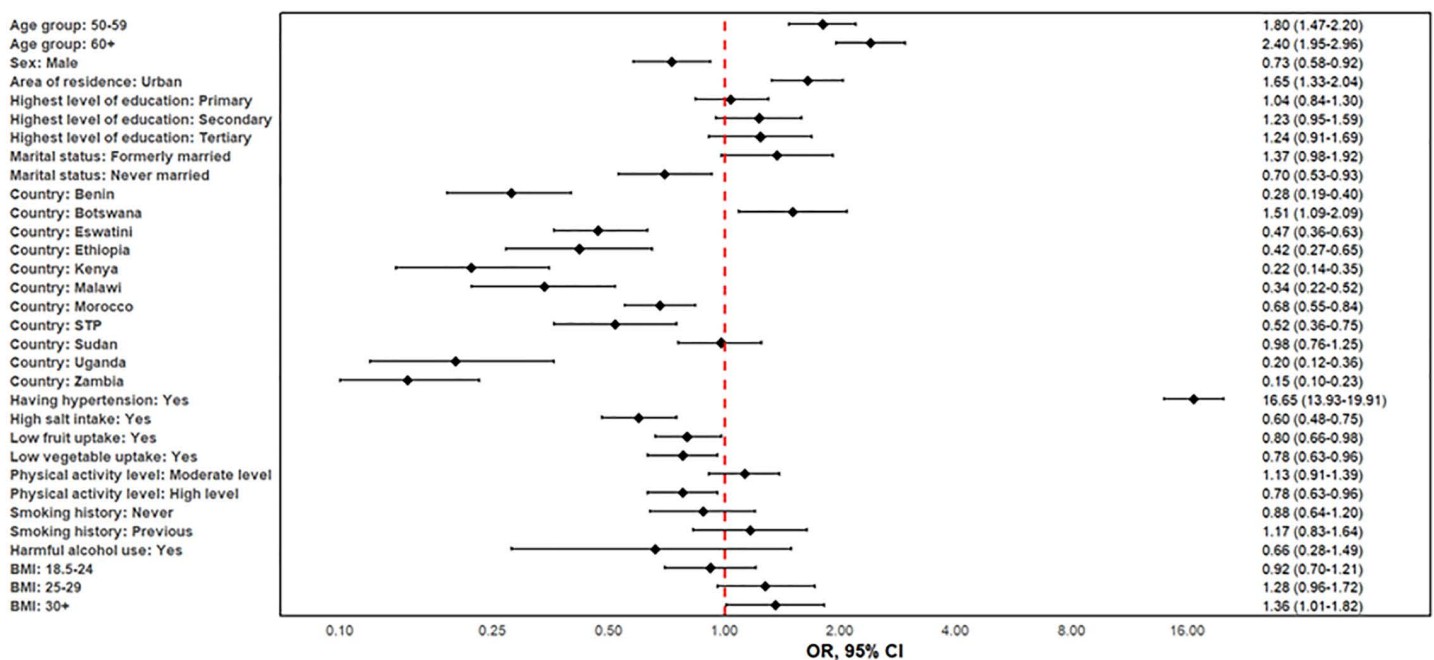

**Fig 4. Factors linked to uptake of preventive cardiovascular disease treatment in twelve African countries, 2014–2019.**

of the uptake of treatment amongst the persons who had CVD is shown in Table 1. Individuals aged below 40 years had lower treatment uptake than individuals aged at least 50 years. The highest treatment uptake was observed amongst those with hypertension (53%), previous smoking history (30%), and obesity (27%) or overweight (40%). The CVD cases in Sudan showed the highest treatment uptake (42%) while the CVD cases in Kenya showed the lowest treatment uptake (5%).

The factors associated with CVD treatment uptake are shown in Fig 5. Age has a major impact on treatment uptake; those in the 50–59 and 60+age groups were far more likely to get therapy than people in the 15–29 age range. For instance, the adjusted model showed a robust correlation between age increased treatment uptake, with an aOR of 2.31 (95%CI:1.39–3.84) for those aged 50–59 and 2.90 (95%CI: 1.75–4.82) for those aged 60+compared to the 15–29-year-old. In the adjusted model, sex did not significantly affect treatment uptake; males had an OR of 1.36 (0.97–1.89) compared to females. However, urban dwellers were more likely than their rural counterparts to receive therapy (OR 1.36, 0.97–1.90). Individuals with hypertension were 7 times more likely to take treatment for CVD than those without hypertension (aOR: 7.19, 95%CI: 5.15–9.78, P<0.001), and individuals with low vegetable uptake were less likely to take CVD treatment than those who had high uptake of vegetables (aOR: 0.58, 95%CI: 0.54–0.83, P=0.003). Kenya, Malawi and Zambia had lower CVD therapy uptake compared to Algeria while Sudan had a similar CVD uptake compared to Algeria. With ORs of 1.37 (0.69–2.74), 0.74 (0.50–1.10) and 1.09 (0.70–1.72), respectively, diabetes, physical inactivity and current smoking demonstrated a weaker or non-significant correlation.

## Discussion

This study examined the prevalence of CVD and the uptake of both preventive and treatment services across twelve African countries, drawing on nationally representative WHO STEPS data. The discussion is organized around three key themes: CVD prevalence, prevention, and treatment uptake. This is followed by an outline of the study's strengths and limitations, and finally the conclusions and recommendations.

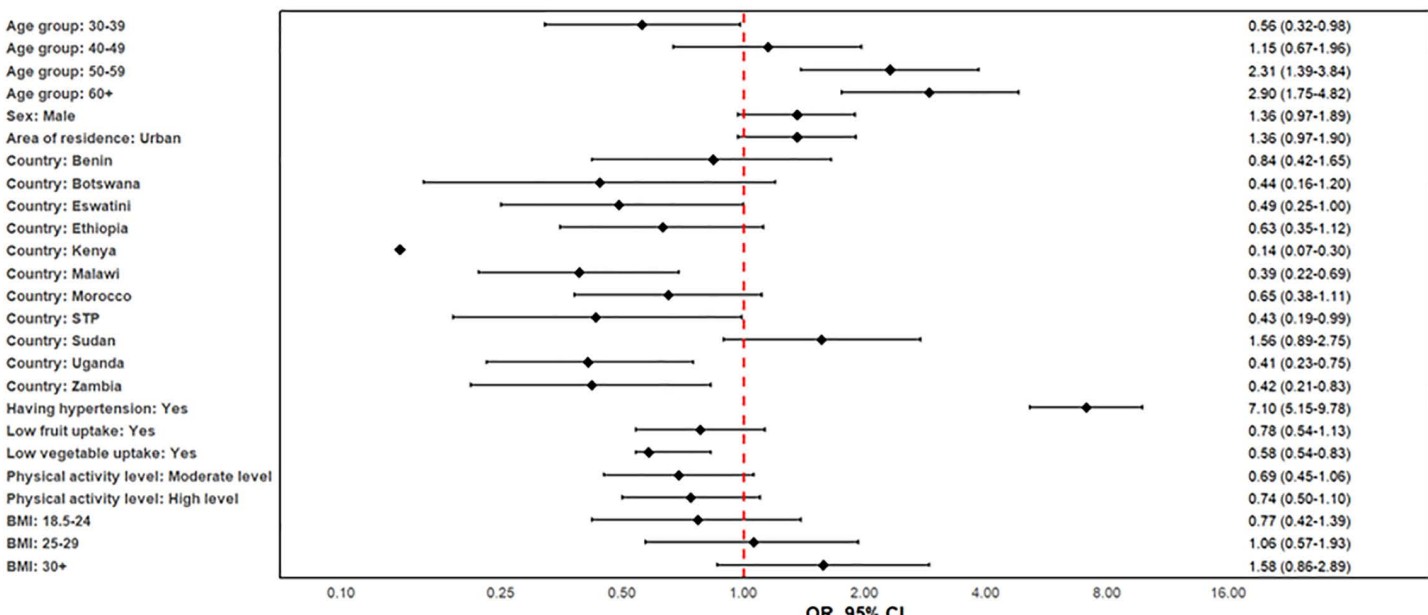

**Fig 5. Factors linked to treatment uptake amongst those diagnosed with cardiovascular disease in twelve African countries, 2014–2019.**

## Discussion for CVD prevalence

The overall CVD prevalence of 5% in this study was consistent with findings from a meta-analysis that found the CVD prevalence to be 7% in SSA [34]. In contrast, other regions exhibit higher CVD prevalence, for example, Europe, South and Southeast Asia and Central Asia have some of the highest CVD mortality rates globally [35,36]. Similarly, North America has a high prevalence of CVD, with projections indicating that 61% of U.S. adults will have some form of cardiovascular disease by 2050 [36]. The low rates of CVDs in Africa may be attributed to limited diagnostic capabilities for confirmation [5]. In addition, the age distribution of the sample, including the exclusion of adults over 69 years, may have contributed to lower estimates, as CVD risk increases with age. Both factors should be considered when interpreting these findings relative to regional prevalence estimates. As people get older, their heart and blood vessels naturally work less efficiently, which greatly increases the risk of heart disease and stroke [37] and this trend was also observed in this study. The implication is to provide the screening services to persons aged 40 years and above, as these are documented to have a high risk of CVD [3,38,39].

Our study is also consistent with what has been reported in other settings that more females had CVD than males [40]. Therefore, CVD prevention may be significantly increased by offering sex-specific risk factor monitoring and intervention strategies [3,40]. High blood pressure is one of the risk factors for CVD that has a high prevalence and is linked to strong evidence of causation [41]. Meta-analyses have shown that lowering blood pressure can effectively prevent CVDs [40,42,43]. Further evidence suggests that smoking is a key factor responsible for more than 30% of the CVD mortality [44,45] [47, pp. 2011–2018]. Therefore, implementing interventions aimed at smoking cessation or preventing exposure to passive smoking has the potential to further reduce the risk of CVD.

Our study found that the prevalence of CVD varies geographically throughout Africa, which emphasises how crucial it is to take regional variances into account when creating focused healthcare interventions. In addition, contextual elements such as the varying stages of national NCD program implementation, differences in the rollout of WHO PEN interventions, and diverse cultural and socioeconomic environments likely contribute to the observed patterns [46]. Incorporating

these broader contextual influences aligns with evidence from other regions, including Europe, where cultural practices, socioeconomic structures, and policy environments shape both CVD prevalence and management. Including these considerations provides a more comprehensive explanation for cross-country variation in our findings. For example, dietary practices, public health policies, and healthcare infrastructure have all been linked to differences in the burden of CVD between Western and Southern European countries [47]. These results highlight the necessity of investigating and addressing the fundamental causes of regional variations in CVD prevalence in order to create efficient, situation-specific management and prevention plans in Africa.

The increased prevalence of CVDs in Africa is largely caused by high salt intake, especially because it is linked to hypertension, a key risk factor for heart disease and stroke. According to a meta-analysis, the World Health Organization's recommended daily limit of 2 grammes of salt is frequently exceeded in Africa, raising the risk of hypertension and cardio-vascular diseases [48]. Regions such as Europe, North America, and Australia have successfully reduced salt consumption through public health measures [49]. For example, in order to prevent hypertension, the American Heart Association recommends avoiding meals high in sodium and focusing on controlling blood pressure through a heart-healthy diet. In Africa, implementing food-based dietary guidelines (FBDGs) has been successful in reducing risk factors for CVD, including high blood pressure and cholesterol. According to studies, choosing a diet that complies with current dietary guidelines decreases blood pressure and cholesterol, which should reduce the risk of CVD by one-third in middle-aged and older people in good health [50]. It has been demonstrated that the Dietary Approaches to Stop Hypertension (DASH) diet, which places an emphasis on consuming less salt, lowers blood pressure, cholesterol, and saturated fats, all of which are risk factors for CVD [51]. In order to prevent premature mortality, lower the increasing burden of non-communicable diseases in Africa, and limit CVD risk factors like hypertension and hypercholesterolemia, it is imperative that high salt intake be addressed through multi-sectoral policies and customised FBDGs.

The CVD were shown to be three times more common among people with hypertension in Africa than in those with normal blood pressure. This pattern is similar to findings from other locations, but the risk varies. Due in large part to improved blood pressure control and easy access to antihypertensive medication, the relative risk of CVD among hypertensive people in Asia, Latin America, North America and Europe is two to three times higher compared with those without hypertension [52,53,54,55]. This emphasises the critical need for better blood pressure screening, reasonably priced antihypertensive drugs, and strong primary healthcare interventions because hypertension is a major modifiable risk factor for CVD. Ignoring this problem will increase the already increasing burden of CVD, worsen health disparities, and put further strain on the region's already vulnerable healthcare infrastructure. Effectively managing hypertension could lower the risk of CVD considerably and help achieve international goals such as the Sustainable Development Goal of lowering premature mortality from non-communicable diseases by 2030.

## Discussion for CVD prevention

Comparing Africa to other regions, the region's 11% adoption rate of CVD preventative therapy is far lower. Due to strong healthcare systems, easily accessible drugs, and effective follow-up procedures, the uptake of preventative medicines like aspirin, statins, and antihypertensives approaches 50%–60% in high-income nations like Europe, North America, and Australia [56,57]. Although acceptance rates in Asia vary, they are typically also high, ranging from 20% to 40%, especially in cities with more advanced healthcare facilities [54,58]. Similar to Africa, Latin America shows also uptake rates of between 30% and 40%, however rural areas encounter similar difficulties [59]. Higher CVD morbidity and mortality, higher healthcare costs as a result of acute care needs, and growing disparities in healthcare access are only a few of the serious ramifications of the poor uptake in Africa [60]. In order to improve CVD preventive care and lower the rising burden of CVD in the region, this finding emphasises the urgent need for targeted interventions like bolstering primary healthcare systems, guaranteeing affordable access to necessary medications, delegating tasks to community health workers, and implementing international initiatives like the WHO's HEARTS program.

According to our study, men were much less likely than women to obtain CVD prevention. This pattern is in accordance with research from other areas, such as Europe and Asia, where women are more likely to follow preventative care recommendations and have higher rates of healthcare utilisation [61]. Because of maternity and reproductive health services, women engage with healthcare systems more frequently, which opens up potential for early CVD risk assessment and intervention. Furthermore, because of cultural norms, a perceived lack of sensitivity to chronic diseases, and poorer health-seeking behavior, men may be less involved in preventive healthcare [62]. This gender gap is made worse in Africa by structural issues including inadequate health care. Systemic issues including low health literacy and resource shortages, especially in rural regions where men predominate, worsen the gender gap in Africa. Public health is significantly impacted by men's lower adoption of CVD prevention. Addressing these inequities is crucial because men in Africa are more likely to suffer from risk factors such smoking, excessive alcohol use, and hypertension [63]. This disparity might be closed and the burden of CVD considerably decreased with gender-specific programs that focus on male health behaviour, raise knowledge, and enhance access to CVD preventive treatment.

Compared to their rural counterparts, urban dwellers had almost double the likelihood of receiving CVD prevention. This gap between urban and rural areas is consistent with international findings showing that urban areas have better access to, infrastructure for, and awareness of, healthcare [64]. Preventive programs that encourage CVD screening and early intervention are more prevalent in urban locations, as are healthcare facilities and specialised physicians. On the other hand, access to preventative treatment is frequently restricted in rural Africa due to factors like lower socioeconomic status, understaffed healthcare facilities, and geographic inaccessibility [65]. Given that the majority of people in Africa live in rural areas, the effects of this urban-rural divide are crucial. Rural dwellers will continue to be disproportionately susceptible to untreated cardiovascular disease risk factors in the absence of focused efforts, which will increase morbidity and mortality. Increasing access to CVD prevention treatment in rural areas through community-based initiatives, mobile health interventions, and decentralised healthcare services must be a top priority for policymakers. For equal progress towards reaching the Sustainable Development Goal of lowering premature mortality from non-communicable diseases by 2030, this gap must be closed.

### Discussion for CVD treatment uptake

There is not much evidence on the uptake of treatment for CVD in Africa. Just 22% of people with CVD in Africa received treatment, indicating a significant gap in the provision of necessary care for those impacted. This number is significantly lower than in other continents like North America and Europe, where more than 70% of people with CVD receive treatment because of their well-established healthcare systems, easy access to drugs, and successful preventative initiatives [66]. Treatment rates range from 40% to 60% across Asia and Latin America, which reflects attempts to scale up the management of chronic diseases and improve healthcare accessibility [67]. The glaring disparity in treatment uptake highlights the persistent issues in Africa, such as the lack of adequate healthcare infrastructure, the high cost of necessary pharmaceuticals, low knowledge, and a paucity of qualified medical personnel, especially in underserved and rural areas [58]. This low treatment rate has serious ramifications because untreated CVDs greatly raises the risk of negative outcomes like heart attack, stroke, and early death. Given the continued challenges with communicable diseases and the increasing burden of CVDs in Africa, region-specific therapies are desperately needed. To close this gap, it is crucial to address socioeconomic barriers, integrate CVD care into basic healthcare, and provide access to cheap treatment.

In Africa, those with hypertension were seven times more likely than those without to receive treatment for CVD, highlighting that hypertension not only serves as a crucial starting point for CVD care but also aids in its diagnosis, given its well-established role as a major risk factor. This pattern is consistent with findings from North America and Europe, where clinical guidelines prioritise screening and treatment for hypertensive people [68,52]. However, missed chances for early detection and management of CVD among other high-risk populations, such as those with diabetes, obesity, or tobacco use, are reflected in the poorer treatment uptake among non-hypertensive adults in Africa [67]. To close this gap and

guarantee fair treatment access for all at-risk persons, thorough risk assessment instruments and integrated care pathways must be expanded. In order to meet global goals like the WHO's "25 by 25" program to reduce early deaths from non-communicable illnesses and lessen the rising burden of CVD in Africa, these measures must be scaled up.

### CVD care cascade

A full understanding of CVD care requires looking at the entire cascade, from screening and diagnosis through to treatment initiation and long-term adherence. The current WHO STEPS dataset does not include information on CVD screening, which limits the ability to assess the full pathway of care. Even so, our analysis shows major gaps: only 11% of participants reported receiving any form of CVD preventive therapy, and just 22% of those with CVD were on treatment. These figures point to large losses at multiple points in the continuum. The gaps are especially evident among rural populations and men, who were less likely to be receiving care [5]. To strengthen future analyses, WHO STEPS surveys should collect data not only on diagnosis and treatment but also on screening and adherence [5]. Alongside this, complementary qualitative work or topic modelling could help explain country-specific barriers, giving clearer insight into how to improve both equity and coverage in CVD care.

To address these gaps, targeted interventions should be prioritized, including integrating CVD care into primary healthcare systems, task-shifting responsibilities to community health workers, and ensuring the availability of low-cost essential medications. When compared to other LMICs, such as countries in Asia and Latin America, Africa's CVD preventive and treatment uptake remains substantially lower, underscoring persistent structural and socioeconomic barriers [5]. Future WHO STEPS iterations could further enhance policy relevance by including detailed care cascade indicators and socio-economic measures, enabling researchers and policymakers to identify critical points for intervention and design context-specific strategies to improve CVD outcomes.

All stages of the CVD cascade (screening, diagnosis, treatment, etc.) depend on people's recall of prior care and on each prior step being completed, so country differences often reflect how many people ever got checked rather than just true disease rates [69]. For example, in Morocco's 2017 survey many older adults may have had routine blood-pressure or cholesterol tests in strong primary care settings and thus report knowing their diagnosis, whereas in Ethiopia's 2015 survey a similar person in a rural area may never have been screened at all. That means Morocco can show a higher percentage "diagnosed" or "treated" simply because more cases were detected, while underdiagnosis in low-resource settings like Malawi or Benin shrinks all subsequent steps. Using self-reported pathways also invites recall and selection biases – for instance, educated or urban patients who seek care are more likely to remain in the cascade – as experts have warned of "recall bias in self-reported information" and poor comparability across surveys [70]. In practice, therefore, differences in cascade figures between countries (e.g., Algeria vs Sudan or Kenya vs Uganda) should be seen as reflecting both epidemiological variation and health-system capacity (screening reach, diagnostic availability), not as direct measures of one country "performing" better than another.

### Strengths and limitations

This is to our knowledge the first multi-country study analyzing prevalence and uptake of CVD treatment in Africa using WHO STEPS. There are some analyses at country level using other sources, but such datasets, unlike WHO STEPS, are not usually nationally representative [5,34]. Therefore, the major strengths of this study were the large sample size and its representativeness, which potentially allows the analysis to improve operational practice and inform policy change. Another strength is that the study covered twelve countries and thus provides a good picture of CVD prevalence and treatment in Africa in general.

However, several limitations should be noted. First, the analysis did not incorporate country-level health system indicators, such as the availability of essential medicines (e.g., aspirin or statins) or the existence of national NCD strategies, which may partly explain inter-country differences. Second, the study relied on self-reported information regarding heart

attacks, CVD treatment, and related risk and protective factors. This approach may introduce diagnostic and recall bias, particularly in settings with limited healthcare access, where individuals may be unaware of or unable to accurately report their condition or treatment history. These factors should be considered when interpreting the results and may provide avenues for future research. We were, however, unable to estimate the extent of underestimation or overestimation of the reported CVD-related risk and protective factors with heart attack and CVD treatment uptake. Although treatment-related analyses incorporated STEPS individual sampling weights, primary sampling units and strata could not be specified due to errors arising from the pooled, harmonized dataset, which may have led to underestimated standard errors and overly narrow confidence intervals, and no formal sensitivity analyses using alternative survey specifications were conducted. Furthermore, the data were available for only one survey round per country, limiting assessment of within-country temporal trends. Although survey years varied across countries, survey year was not significantly associated with CVD risk; however, residual temporal heterogeneity may still affect between-country comparisons.

The study only looked at the general CVD conditions and does not provide information on the type of CVD that the individuals had. Having such information could help to choose the most suitable risk reduction interventions. Another limitation is that the WHO STEPS does not capture information on screening for CVDs although such information would better help understand the care cascade from those eligible for screening to uptake of treatment. Besides, wealth quintile is not captured in most of the WHO STEPS surveys despite wealth index being paramount in determining equity and inequity of health outcomes. For example, there is evidence from multiple countries across the world that wealthier individuals had higher prevalence of CVD than poorer ones [71,72].

Although wealth quintile data were unavailable, urban/rural residence, gender, and education were used as proxies to explore socioeconomic gradients in CVD prevention and treatment uptake. Urban residents and women were more likely to receive care, reflecting disparities in healthcare access and health-seeking behavior. While country-specific differences (e.g., Uganda vs. Sudan) were observed, they were not fully contextualized. Future work could explore the underlying reasons such as healthcare access, survey methodology, or cultural factors using qualitative approaches or text-based analyses like topic modeling to better understand country-level patterns.

## Conclusion

In conclusion; using a sizable and representative dataset from twelve countries, our work offers a crucial insight into the prevalence of CVDs and the uptake of its treatment in Africa. The African context highlights gaps in diagnosis and treatment for CVD compared to Europe, Asia, and North America. People with hypertension are seven times more likely to undergo treatment, making hypertension a crucial point of entry for CVD therapy. Important risk factors, such as smoking and excessive salt consumption, highlight the need for multi-sectoral treatments like smoking cessation programs and nutritional recommendations. Notwithstanding its advantages, such as its sizable sample size and representativeness, the study's dependence on self-reported data and its absence of details on wealth indices and particular forms of CVDs point to areas that require more investigation. Achieving global health goals like the Sustainable Development Goals and lowering the rising burden of CVDs in Africa depend on closing these disparities and putting equitable, evidence-based interventions into place.

## Supporting information

**S1 File.** *1_Create_dataset_for_analysis.* *R* R script used to generate the dataset for analysis.
(ZIP)

**S2 File. Sample simulation code.**
(DOCX)

## Acknowledgments

The authors would like to thank the World Health Organisation for granting access to use the WHO STEPS data sets for Africa.

## Author contributions

**Conceptualization:** Wingston Felix Ng'ambi, Janne Estill, Fatma Aziza Merzouki, Cosamas Zyambo, Jonathan Chiwanda Banda.

**Data curation:** Wingston Felix Ng'ambi.

**Formal analysis:** Wingston Felix Ng'ambi.

**Investigation:** Olivia Keiser.

**Methodology:** Wingston Felix Ng'ambi, Janne Estill, Jonathan Chiwanda Banda, David Beran, Olivia Keiser.

**Project administration:** Wingston Felix Ng'ambi.

**Resources:** Wingston Felix Ng'ambi.

**Software:** Wingston Felix Ng'ambi.

**Supervision:** Wingston Felix Ng'ambi, Janne Estill, Fatma Aziza Merzouki, Olivia Keiser.

**Validation:** Wingston Felix Ng'ambi, Jonathan Chiwanda Banda, David Beran, Olivia Keiser.

**Visualization:** Wingston Felix Ng'ambi.

**Writing – original draft:** Wingston Felix Ng'ambi.

**Writing – review & editing:** Wingston Felix Ng'ambi, Janne Estill, Fatma Aziza Merzouki, Cosamas Zyambo, Jonathan Chiwanda Banda, David Beran, Olivia Keiser.

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
