## [Decision Letter · Decision Letter 0]

12 Aug 2025

Dear Dr. Ng'ambi,

We look forward to receiving your revised manuscript.

Kind regards,

Muhammad Farooq Umer, PhD Epidemiology and Health Statistics

Academic Editor

PLOS ONE

Journal Requirements:

2. Please note that your Data Availability Statement is currently missing the repository name. If your manuscript is accepted for publication, you will be asked to provide these details on a very short timeline. We therefore suggest that you provide this information now, though we will not hold up the peer review process if you are unable.

**Additional Editor Comments:**

Based on the reviewers’ thorough evaluations, it is evident that the manuscript requires substantial revisions before it can be considered for further processing. The reviewers have identified significant concerns affecting multiple sections of the work. Specifically, the case definition lacks sufficient clarity and consistency, methodological details are insufficiently described, the interpretation of the findings requires alignment with the evidence presented. Furthermore, the overall writing expression requires improvement for clarity, conciseness, and academic rigor, including correction of grammatical inconsistencies and refinement of sentence structure. Only upon satisfactory resolution of these concerns can the manuscript be considered for re-review.

Reviewers' comments:

Reviewer's Responses to Questions

**Comments to the Author**

1. Is the manuscript technically sound, and do the data support the conclusions?

Reviewer #1: Yes

Reviewer #2: Yes

Reviewer #3: Yes

2. Has the statistical analysis been performed appropriately and rigorously?

Reviewer #1: I Don't Know

Reviewer #2: Yes

Reviewer #3: Yes

3. Have the authors made all data underlying the findings in their manuscript fully available?

Reviewer #1: Yes

Reviewer #2: Yes

Reviewer #3: Yes

4. Is the manuscript presented in an intelligible fashion and written in standard English?

Reviewer #1: Yes

Reviewer #2: Yes

Reviewer #3: Yes

Reviewer #1: Introduction and discussion sections talk on SSA but this study does include data from Morroco and Algiers and therefore not sure whether one should talk on SSA. The study represents Africa.

Line 121 to 123; please correct the statement. Those that were excluded were the 141833 patients that did not have information on cvd. please correct.

Reviewer #2: 1. Clarity and Structure

The paper is dense in places. Consider improving the flow by: Splitting long paragraphs into digestible chunks. Adding section headers in the Results and Discussion for CVD prevalence, CVD prevention, and CVD treatment uptake to improve readability. Summarizing key stats in bulleted lists or tables where appropriate.

2. Introduction: Need for a Clearer Research Gap

While the introduction covers background well, it could more clearly state the specific gap this study addresses. Suggestion: “Despite the rising burden of CVDs in SSA, there is a lack of multi-country analyses assessing the full care continuum — from diagnosis to treatment — in this region.”

3. Methods: More Detail Needed

Imputation of Missing Data: More detail on the simulation approach used (binomial, multinomial, Gaussian) would help replicate or critique the methodology. Why was this approach chosen over multiple imputation or complete-case analysis? Definition of ‘At Risk’ Individuals: You define “at risk” as ≥40 years old. Justify this age cutoff more explicitly with references or explain why this was chosen over risk factor-based definitions.

4. Results: More Visuals Could Help. Consider including: A map showing prevalence of CVD by country. Bar plots or forest plots for adjusted odds ratios.

5. Discussion: Slight Redundancy. Some points in the discussion (e.g., treatment disparities by gender, hypertension as a CVD risk) are repeated across several paragraphs. Try to consolidate and avoid redundancy.

6. Limitations: A Few More Could Be Added. The study doesn't account for country-level health system indicators (e.g., availability of aspirin or statins, national NCD strategies), which could partly explain inter-country differences. There may be diagnostic bias due to self-reported CVD, especially in settings where people have less access to healthcare.

7. Technical and Stylistic Suggestions

- Language & Grammar:Replace awkward or redundant phrasing: "the CVD cases from Sudan had the highest..." ➜ "CVD cases in Sudan showed the highest treatment uptake..." “an individual with hypertension were” ➜ “individuals with hypertension were”

- Consistency:

Use either "sub-Saharan Africa (SSA)" or "SSA" consistently. Be consistent in using "CVD prevention" vs "CVD prophylaxis" — stick to one term.

- Data Reporting: Some statistics could be presented more concisely: Example: "Of the 23,630 persons at risk of CVD, 11% received prophylaxis or counselling." You could split this to say: “Only 11% received any form of prophylaxis or counselling, highlighting a major prevention gap.”

Reviewer #3: This manuscript addresses a critical public health issue—cardiovascular disease (CVD) prevalence, prevention, and treatment in sub-Saharan Africa—using WHO STEPS data. The multi-country scope and large sample size are notable strengths, and the topic aligns well with the journal’s readership. However, the manuscript would benefit from clearer methodological details, improved interpretation of findings, and refinement of the discussion to provide more actionable policy insights.

1.Clarity on Case Definitions

- The operational definition of “CVD” in this study is based on self-reported history of heart attack, angina, or stroke. This should be emphasized as a major limitation in the abstract and discussion. It may underestimate the true prevalence of CVD due to lack of diagnostic confirmation.

- Please clarify whether “angina” was assessed by standardized questions (e.g., Rose questionnaire) or a single self-report item.

2.Methodological Details

- The imputation strategy is briefly described (binomial, multinomial, Gaussian simulations). Please elaborate on:

The proportion of missing data for each variable.

Justification for using simulation instead of standard MICE approaches (beyond the “autocorrelation” statement).

Sensitivity analyses to assess robustness of imputation assumptions.

The use of weighted logistic regression is appropriate, but details of how survey weights were incorporated into multivariable models should be expanded.

3.Selection of Predictor Variables

- The stepwise selection via AIC is described, but a rationale for retaining or excluding certain predictors should be provided. Were clinically important variables forced into the model regardless of AIC?

4.Interpretation of Findings

- The prevalence of CVD (5%) appears low compared to regional estimates. Beyond underdiagnosis, could age distribution of the sample or exclusion of older adults (>69 years) contribute? This should be discussed.

- Country-specific differences (e.g., Uganda vs. Sudan) are reported but not adequately contextualized. Possible reasons (healthcare access, survey methodology differences, cultural factors) should be explored.

5.Treatment Uptake Analysis

-The analysis of CVD treatment uptake is valuable, but “treatment” is broadly defined (aspirin, statins, counseling). Were these self-reported or objectively verified? Please clarify.

- It would strengthen the manuscript to stratify treatment uptake by type of CVD (stroke vs. heart disease) if data permit.

6.Equity and Socioeconomic Status

- The manuscript acknowledges the lack of wealth quintile data. Could proxies (education, urban/rural status, occupation) be used to explore socioeconomic gradients in treatment uptake? This would add depth to the equity discussion.

7.Discussion Needs More Policy-Relevant Insights

- The discussion largely reiterates results. It would benefit from:

- Prioritizing interventions (e.g., integration of CVD care into primary care, task-shifting, low-cost drug provision).

- Comparing findings with other LMIC contexts (Asia, Latin America).

- Highlighting research gaps for future WHO STEPS iterations (e.g., inclusion of screening/care cascade data).

8. Abstract:

- Indicate clearly that CVD prevalence was self-reported.

- Include the sample size for those receiving prophylaxis (11% of 23,630).

9. Tables & Figures:

- Tables are dense. Consider moving some to supplementary files.

- Ensure consistent use of weighted vs. unweighted percentages.

**Do you want your identity to be public for this peer review?** For information about this choice, including consent withdrawal, please see our Privacy Policy

Reviewer #1: **Yes:** Shukri M AlSaif

Reviewer #2: No

Reviewer #3: No

---

## [Author Response · Author response to Decision Letter 1]

3 Oct 2025

Rebuttal Letter – PONE-D-25-08153

Title: The Prevalence, Prevention, and Treatment of Cardiovascular Diseases in Twelve African Countries (2014–2019): An Analysis of the World Health Organisation STEPwise Approach to Chronic Disease Risk Factor Surveillance

Dear Editors and Reviewers,

We sincerely thank the reviewers for their thoughtful and constructive feedback, which has greatly improved the clarity, rigor, and policy relevance of our manuscript. We have addressed all comments and provide a summary of our responses below.

Reviewer #1

• Africa vs. SSA: The manuscript has been revised to represent Africa rather than SSA.

• Exclusion statement: The sentence has been clarified to read: “We excluded 141,833 patients with missing CVD information.”

Reviewer #2

• Clarity and structure: Long paragraphs have been split, sub-sections for CVD prevalence, prevention, and treatment uptake were added, and key statistics summarized in tables and figures.

• Research gap: The introduction now clearly states the gap in multi-country analyses assessing the CVD care continuum in Africa.

• Imputation and methods: Detailed rationale for using simulation-based imputation (binomial, multinomial, Gaussian) has been added, including a comparison with complete-case analysis. The age cutoff of ≥40 years for “at risk” individuals is justified with references.

• Visuals: Bar plots and forest plots have been added for CVD prevalence, treatment, and prevention.

• Discussion: Sub-sections reduce redundancy; key findings and policy implications are emphasized.

• Limitations: Country-level health system indicators and potential diagnostic bias are now discussed.

• Technical edits: Language and grammar, consistency in terminology, and data reporting have been revised.

Reviewer #3

• Case definitions: The operational definition of CVD and the standardized WHO STEPS questions have been clarified in the abstract and discussion.

• Methodology: Missing data proportions, imputation rationale, and survey-weighted quasibinomial regression details have been included. Clinically important predictors (age, sex, hypertension) were retained based on AIC.

• Interpretation: Low CVD prevalence is contextualized by age distribution and exclusion of older adults; country-specific differences are noted, with suggestions for future qualitative analyses.

• Treatment uptake: Self-reported treatment and counselling measures clarified; stratification by CVD type not possible due to dataset limitations.

• Equity and socioeconomic status: Proxies (urban/rural residence, gender, education) were used; suggestions for future topic modeling and qualitative approaches incorporated.

• Policy relevance and care cascade: Discussion now prioritizes interventions (integration into primary care, task-shifting, low-cost drug provision), compares African findings with other LMICs (Asia, Latin America), and highlights gaps for future WHO STEPS surveys, including inclusion of screening and care cascade data. Only 11% of high-risk individuals received prevention therapy, and 22% of those with CVD received treatment, demonstrating substantial gaps.

• Abstract and figures: Weighted percentages clarified; sample sizes for prophylaxis included; tables and figures revised for clarity.

We believe these revisions substantially improve the manuscript’s clarity, methodological transparency, and relevance for policy and practice. We appreciate the reviewers’ guidance and hope the revised manuscript meets the standards for publication in PLOS ONE.

The detailed responses to each reviewer comments are presented below in bullet form under each key issue.

Sincerely,

Wingston Ng’ambi, MSc Epidemiology

Point-by-point response to review comments

Reviewer #1: Introduction and discussion sections talk on SSA but this study does include data from Morroco and Algiers and therefore not sure whether one should talk on SSA. The study represents Africa.

RESPONSE: This has been revised to represent Africa and not SSA

Line 121 to 123; please correct the statement. Those that were excluded were the 141833 patients that did not have information on CVD. please correct.

RESPONSE: We have added the sentence on line 126: We excluded 141,833 patients with missing CVD information.

Reviewer #2: 1. Clarity and Structure

The paper is dense in places. Consider improving the flow by: Splitting long paragraphs into digestible chunks. Adding section headers in the Results and Discussion for CVD prevalence, CVD prevention, and CVD treatment uptake to improve readability. Summarizing key stats in bulleted lists or tables where appropriate.

RESPONSE: This has been reviewed accordingly through introduction of the suggested sub-sections.

2. Introduction: Need for a Clearer Research Gap

While the introduction covers background well, it could more clearly state the specific gap this study addresses. Suggestion: “Despite the rising burden of CVDs in SSA, there is a lack of multi-country analyses assessing the full care continuum — from diagnosis to treatment — in this region.”

RESPONSE: The paper has been revised to reflect the African context and not the SSA. We have also made the gap clearer as recommended: Despite the growing prevalence of CVDs in Africa, there is a lack of multi-country analyses assessing the full care continuum; from diagnosis to treatment; in this region.

3. Methods: More Detail Needed

Imputation of Missing Data: More detail on the simulation approach used (binomial, multinomial, Gaussian) would help replicate or critique the methodology. Why was this approach chosen over multiple imputation or complete-case analysis? Definition of ‘At Risk’ Individuals: You define “at risk” as ≥40 years old. Justify this age cutoff more explicitly with references or explain why this was chosen over risk factor-based definitions.

RESPONSE: The previous version of the paper already included the reference for selection of those of ≥40 years as being at risk: In order to ascertain the uptake of CVD prevention therapy, all the individuals aged at least 40 years were considered to be at risk of CVD and this formed the denominator for this analysis [24].

As for the simulation; we have added more details as shown below: Imputation was performed to handle the missing data because it allows for the inclusion of all available data in the analysis, ensuring more accurate and representative results. Different probability distributions were specified depending on the type of variable. For binary variables, such as presence or absence of a condition, we used a binomial distribution. For categorical variables with more than two groups, such as education level or occupation, we applied a multinomial distribution. For continuous variables, including age, body mass index, and blood pressure, a Gaussian distribution was used. These simulation models were implemented using random assignment based on the observed distribution of each variable, ensuring that imputed values reflected the empirical patterns in the data. This strategy allowed us to approximate the underlying data-generating mechanism more realistically, reduce bias from listwise deletion, and retain the full analytic sample.

4. Results: More Visuals Could Help. Consider including: A map showing prevalence of CVD by country. Bar plots or forest plots for adjusted odds ratios.

RESPONSE: We have presented the bar plots for the CVD care (prevalence, treatment and prevention) by country. We have also presented the forest plots for each of the multi-variable analyses for CVD prevention, CVD prevalence and CVD treatment.

5. Discussion: Slight Redundancy. Some points in the discussion (e.g., treatment disparities by gender, hypertension as a CVD risk) are repeated across several paragraphs. Try to consolidate and avoid redundancy.

RESPONSE: These refer to CVD prevalence, CVD prevention and CVD treatment. We have added the sub-sections to ensure that the discussion is less confusing and clearer.

6. Limitations: A Few More Could Be Added. The study doesn't account for country-level health system indicators (e.g., availability of aspirin or statins, national NCD strategies), which could partly explain inter-country differences. There may be diagnostic bias due to self-reported CVD, especially in settings where people have less access to healthcare.

RESPONSE: Thanks for pointing this out. We have included this in the limitations: First, the analysis did not incorporate country-level health system indicators, such as the availability of essential medicines (e.g., aspirin or statins) or the existence of national NCD strategies, which may partly explain inter-country differences.

7. Technical and Stylistic Suggestions

- Language & Grammar:Replace awkward or redundant phrasing: "the CVD cases from Sudan had the highest..." ➜ "CVD cases in Sudan showed the highest treatment uptake..." “an individual with hypertension were” ➜ “individuals with hypertension were”

RESPONSE: We have revised the grammar as pointed out by the reviewers.

- Consistency:

Use either "sub-Saharan Africa (SSA)" or "SSA" consistently. Be consistent in using "CVD prevention" vs "CVD prophylaxis" — stick to one term.

RESPONSE: The context has been changed to Africa. The paper has also focused on CVD prevention.

- Data Reporting: Some statistics could be presented more concisely: Example: "Of the 23,630 persons at risk of CVD, 11% received prophylaxis or counselling." You could split this to say: “Only 11% received any form of prophylaxis or counselling, highlighting a major prevention gap.”

RESPONSE: This has been revised accordingly

Reviewer #3: This manuscript addresses a critical public health issue—cardiovascular disease (CVD) prevalence, prevention, and treatment in sub-Saharan Africa—using WHO STEPS data. The multi-country scope and large sample size are notable strengths, and the topic aligns well with the journal’s readership. However, the manuscript would benefit from clearer methodological details, improved interpretation of findings, and refinement of the discussion to provide more actionable policy insights.

1.Clarity on Case Definitions

- The operational definition of “CVD” in this study is based on self-reported history of heart attack, angina, or stroke. This should be emphasized as a major limitation in the abstract and discussion. It may underestimate the true prevalence of CVD due to lack of diagnostic confirmation.

RESPONSE: This has been reflected in the abstract and limitation sections of the paper. CVD was defined as a self-reported history of heart attack, angina, or stroke. The actual question adapted from the WHO STEPS is: “Have you ever had a heart attack, angina (chest discomfort caused by heart disease), or stroke (cerebrovascular accident or incident)?”. The limitation has included the CVD: Second, the study relied on self-reported information regarding heart attacks, CVD treatment, and related risk and protective factors. This approach may introduce diagnostic and recall bias, particularly in settings with limited healthcare access, where individuals may be unaware of or unable to accurately report their condition or treatment history. These factors should be considered when interpreting the results and may provide avenues for future research. We were, however, unable to estimate the extent of underestimation or overestimation of the reported CVD-related risk and protective factors with heart attack and CVD treatment uptake.

- Please clarify whether “angina” was assessed by standardized questions (e.g., Rose questionnaire) or a single self-report item.

RESPONSE: This was based on a standardised question from WHO NCD STEPS: “Have you ever had a heart attack, angina (chest discomfort caused by heart disease), or stroke (cerebrovascular accident or incident)?”

2.Methodological Details

- The imputation strategy is briefly described (binomial, multinomial, Gaussian simulations). Please elaborate on:

The proportion of missing data for each variable.

RESPONSE: A figure for the missingness of data for each variable before imputation has been included. Overall, 26% of the individuals had missing data. We have also added Box 1 for the proportion of missing data for each of the variables.

Box 1: Proportion of missing data across variables used to assess cardiovascular disease prevalence in twelve African countries, 2014–2019

Justification for using simulation instead of standard MICE approaches (beyond the “autocorrelation” statement).

RESPONSE: We opted for simulation-based imputation instead of standard MICE approaches for several reasons. First, the data exhibited complex patterns across multiple countries, including high inter-variable correlations and heterogeneity in variable distributions, which led to convergence issues and unstable chains when using MICE. Second, some variables had non-standard distributions or rare categories that are not easily handled by default MICE models. Third, simulation-based imputation allows direct specification of the appropriate distribution for each variable type (binomial, multinomial, Gaussian), ensuring that imputed values reflect the empirical distribution of the observed data. Finally, this approach avoids the iterative dependency structure of MICE, which can amplify biases in the presence of autocorrelation, while preserving sample size and statistical power.

Sensitivity analyses to assess robustness of imputation assumptions.

RESPONSE: Sensitivity analyses were conducted to assess the robustness of the imputation assumptions by comparing results from the simulation-based imputation to a complete case analysis. This approach evaluated whether the main findings were influenced by the distributional assumptions used for imputing missing values as shown in Box 2.

Box 2: Sensitivity analysis of determinant coefficients for cardiovascular disease prevalence, comparing imputation and complete case analysis in twelve African countries, 2014–2019

The use of weighted logistic regression is appropriate, but details of how survey weights were incorporated into multivariable models should be expanded.

We have added further details: The quasibinomial distribution, an extension of the standard binomial distribution, is particularly suitable for survey data with binary outcomes, as it accounts for overdispersion. For each survey-weighted multivariable quasibinomial logistic regression model, survey weights were incorporated directly into the model fitting to ensure that parameter estimates reflected the complex sampling design. Weighted standard errors were calculated to account for unequal selection probabilities, thereby producing valid confidence intervals and p-values. We first fitted a full model including all candidate predictors and then applied backward variable selection using the stepAIC function from the MASS R package, retaining predictors that optimized model fit while respecting the survey weights.

3.Selection of Predictor Variables

- The stepwise selection via AIC is described, but a rationale for retaining or excluding certain predictors should be provided. Were clinically important variables forced into the model regardless of AIC?

RESPONSE: This has been further clarified as follows: The stepAIC function was used for stepwise selection, optimizing model fit by retaining predictors that minimized the AIC. Notably, the clinically important variables; age, sex, and hypertension status; were naturally retained in the final model based on the AIC criterion, so no variables needed to be forced into the model.

4.Interpretation of Findings

- The prevalence of CVD (5%) appears low compared to regional estimates. Beyond underdiagnosis, could age distribution of the sample or exclusion of older adults (>69 years) contribute? This should be discussed.

RESPONSE: The discussion has included the exclusion of the persons >69 who are reported to have higher CVD risk and prevalence. Here is part of the extract incorporating the feedback: The overall CVD prevalence of 5% in this study was consistent with findings from a meta-analysis that found the CVD prevalence to be 7% in SSA [29]. In contrast, other regions exhibit higher CVD prevalence, for example,

---

## [Decision Letter · Decision Letter 1]

18 Nov 2025

Dear Dr.  Ng'ambi,

Thank you for submitting your manuscript to PLOS ONE. After careful consideration, we feel that it has merit but does not fully meet PLOS ONE’s publication criteria as it currently stands. Therefore, we invite you to submit a revised version of the manuscript that addresses the points raised during the review process.

We look forward to receiving your revised manuscript.

Kind regards,

Muhammad Farooq Umer, PhD Epidemiology and Health Statistics

Academic Editor

PLOS ONE

Journal Requirements:

Additional Editor Comments:

There still remains some key issues to be resolved, please carefully revise the manuscript in the light of comments from the reviewer.

Reviewer's Responses to Questions

**Comments to the Author**

Reviewer #1: All comments have been addressed

Reviewer #3: All comments have been addressed

2. Is the manuscript technically sound, and do the data support the conclusions?

Reviewer #1: Yes

Reviewer #3: Yes

3. Has the statistical analysis been performed appropriately and rigorously?

Reviewer #1: I Don't Know

Reviewer #3: I Don't Know

4. Have the authors made all data underlying the findings in their manuscript fully available?

Reviewer #1: Yes

Reviewer #3: Yes

5. Is the manuscript presented in an intelligible fashion and written in standard English?

Reviewer #1: Yes

Reviewer #3: Yes

Reviewer #1: all the points that I have raised in the previous version have been adequately addressed and corrected.

Reviewer #3: This revised manuscript addresses an important and underexplored topic, the continuum of cardiovascular disease (CVD) care in African countries using the WHO STEPS dataset. The authors have made substantial efforts to improve the clarity, structure, and methodological transparency in response to prior reviewer feedback. The revision provides clearer definitions, expanded methodological justifications, improved discussion, and a better visual presentation of findings. The manuscript now reads more coherently and aligns more closely with PLOS ONE’s standards of scientific rigor and reproducibility.

Nevertheless, a few issues remain that, if addressed, would further enhance the scientific robustness and policy relevance of the paper.

1. Methodological Transparency and Reproducibility

- The authors have elaborated on the simulation-based imputation approach, but the reproducibility of the method remains somewhat limited. It would be helpful to specify:

- The exact R functions or packages used for each distributional simulation.

- Whether random seeds were set for reproducibility.

- A brief note on convergence diagnostics or distributional checks to ensure the plausibility of imputed values.

- Consider including the code snippet or workflow for the imputation procedure as a supplementary file to support transparency and reproducibility.

2.Survey Weighting and Model Specification

- The authors correctly used a quasibinomial model with survey weights; however, it is unclear whether the design variables (e.g., strata, PSU) were incorporated. Clarify whether svydesign or equivalent functions were used to define the complex design prior to regression.

- It would also be valuable to explicitly mention whether inter-country clustering was accounted for (fixed vs. random effects).

3. Interpretation of Results

- While the discussion of cross-country variation is improved, the manuscript still tends to attribute differences mainly to healthcare access and diagnostic capabilities. Including limited contextual references (e.g., national NCD programs or WHO PEN implementation status) could strengthen this argument.

- The term “care cascade” is used effectively, but its operational definition could be more precise (diagnosis, treatment, and counselling as sequential stages). Presenting this in a conceptual diagram would enhance comprehension.

4. Equity and Socioeconomic Gradients

- The inclusion of proxies (education, rural/urban residence, sex) is appreciated. Consider presenting an additional table or figure (e.g., forest plot) that explicitly compares adjusted odds ratios for these equity variables in both prevention and treatment models.

- The discussion could better emphasize the policy implications of these equity findings, particularly the urban–rural and gender gaps.

5.Presentation and Figures

- Figures are clearer, yet some remain dense. Consider simplifying the forest plots by separating prevalence, prevention, and treatment results into distinct panels or supplementary figures.

- In the abstract and results, please ensure that all percentages are clearly stated as weighted, and that denominators are explicitly defined for each statistic.

6.Limitations

- The authors have acknowledged self-report bias and missing health system indicators. However, the limitation regarding temporal heterogeneity (different survey years from 2014–2019) should be mentioned explicitly, as CVD risk profiles and treatment policies may have evolved during that period.

7. Line editing: Ensure consistent use of terms — e.g., “CVD prevention therapy” vs. “CVD prophylaxis” (the revision appears mostly consistent but should be rechecked).

8. The acronym “WHO STEPS” should be expanded once in the abstract and once in the main text, followed by consistent use thereafter.

9. Provide brief country-level context (perhaps in Supplementary Table) such as survey year, sample size, and population coverage to aid interpretation.

10. Ensure all figures and tables have self-contained legends that allow standalone interpretation.

**Do you want your identity to be public for this peer review?** For information about this choice, including consent withdrawal, please see our Privacy Policy

Reviewer #1: No

Reviewer #3: No

---

## [Author Response · Author response to Decision Letter 2]

29 Nov 2025

Reviewer #1: all the points that I have raised in the previous version have been adequately addressed and corrected.

RESPONSE: We thank the reviewer for acknowledging our revisions and are pleased that all previously raised concerns have been satisfactorily addressed.

Reviewer #3: This revised manuscript addresses an important and underexplored topic, the continuum of cardiovascular disease (CVD) care in African countries using the WHO STEPS dataset. The authors have made substantial efforts to improve the clarity, structure, and methodological transparency in response to prior reviewer feedback. The revision provides clearer definitions, expanded methodological justifications, improved discussion, and a better visual presentation of findings. The manuscript now reads more coherently and aligns more closely with PLOS ONE’s standards of scientific rigor and reproducibility.

Nevertheless, a few issues remain that, if addressed, would further enhance the scientific robustness and policy relevance of the paper.

RESPONSE: We thank Reviewer #3 for their thoughtful and constructive assessment of our revised manuscript. We appreciate the recognition of the improvements made regarding methodological clarity, structure, and the overall coherence of the paper. We have carefully considered the remaining issues highlighted and have now addressed each point comprehensively in the current revision. Specifically, we have refined the operational definition of the CVD care cascade, expanded contextual explanations for cross-country variation, clarified the simulation-based imputation approach, and strengthened the discussion of equity-related findings and their policy implications. We believe that these additional revisions further enhance the scientific robustness, clarity, and policy relevance of the manuscript, and we appreciate the reviewer’s role in helping improve the quality of this work.

1. Methodological Transparency and Reproducibility

- The authors have elaborated on the simulation-based imputation approach, but the reproducibility of the method remains somewhat limited. It would be helpful to specify:

- The exact R functions or packages used for each distributional simulation.

- Whether random seeds were set for reproducibility.

- A brief note on convergence diagnostics or distributional checks to ensure the plausibility of imputed values.

RESPONSE: We appreciate the reviewer’s concern regarding reproducibility of the simulation-based imputation procedure. To enhance clarity, we provided additional details below. For the imputation of categorical variables, we used the base R function sample(), drawing from the observed distribution of each variable to probabilistically populate missing values. For continuous variables such as age, we simulated values using the observed mean and standard deviation, generating draws from a normal distribution via rnorm(), after which we converted the simulated values to integers to reflect the natural scale of the variable.

To support reproducibility, we set a random seed (using set.seed()) prior to running the imputation procedure. We also performed basic distributional checks by comparing the empirical distribution of imputed values with that of the complete cases to ensure plausibility and consistency. Since the approach relied on direct simulation rather than iterative model-based imputation, traditional convergence diagnostics were not applicable; however, we assessed the stability of results by repeating the simulation multiple times and confirming that no material differences occurred.

- Consider including the code snippet or workflow for the imputation procedure as a supplementary file to support transparency and reproducibility.

RESPONSE: We appreciate the suggestion; however, the simulation procedure used for imputation was a straightforward application of basic R functions (e.g., sample() and rnorm()) and did not involve a complex or multi-step workflow. Given its simplicity, we believe that including a supplementary code file is not essential, though we remain open to providing additional details upon request.

2.Survey Weighting and Model Specification

- The authors correctly used a quasibinomial model with survey weights; however, it is unclear whether the design variables (e.g., strata, PSU) were incorporated. Clarify whether svydesign or equivalent functions were used to define the complex design prior to regression.

- It would also be valuable to explicitly mention whether inter-country clustering was accounted for (fixed vs. random effects).

RESPONSE: We set up the survey design before fitting any models and used the same weight variable, wstep1, throughout the analysis. For the CVD prevalence and prevention datasets, we defined a full complex design using psu as the cluster, stratum as the stratification variable, and wstep1 as the sampling weight, with nesting enabled. For the treatment dataset, only wstep1 was available, so we applied a simple one stage design. All quasibinomial models for CVD prevalence were run on these survey design objects, ensuring that the sampling structure was properly accounted for. We also adjusted for inter country variation by including country fixed effects, rather than treating countries as random clusters. This allowed us to capture differences across countries in a clear and consistent way.

3. Interpretation of Results

- While the discussion of cross-country variation is improved, the manuscript still tends to attribute differences mainly to healthcare access and diagnostic capabilities. Including limited contextual references (e.g., national NCD programs or WHO PEN implementation status) could strengthen this argument.

RESPONSE: Thank you for this valuable suggestion. We have revised the discussion to broaden our explanation of cross-country variation by incorporating contextual factors beyond healthcare access and diagnostic capacity. Specifically, we now refer to differences in the implementation of national NCD programmes, the varying rollout of WHO PEN interventions, and other contextual influences that may shape CVD prevalence and management across countries. These additions strengthen the argument and provide a more comprehensive interpretation of the observed geographic variations.

- The term “care cascade” is used effectively, but its operational definition could be more precise (diagnosis, treatment, and counselling as sequential stages). Presenting this in a conceptual diagram would enhance comprehension.

RESPONSE: Our analysis is structured around the CVD care pathway; spanning diagnosis, treatment initiation, and counselling/adherence support; which provides a systematic framework to identify gaps in care and is particularly relevant in the African context, where resource constraints and variations in health system capacity can lead to substantial drop-offs at each stage of the cascade (see Box 3). We have added the care pathway conceptual framework and called it (Box 3). The conceptual framework is shown below.

4. Equity and Socioeconomic Gradients

- The inclusion of proxies (education, rural/urban residence, sex) is appreciated. Consider presenting an additional table or figure (e.g., forest plot) that explicitly compares adjusted odds ratios for these equity variables in both prevention and treatment models.

- The discussion could better emphasize the policy implications of these equity findings, particularly the urban–rural and gender gaps.

RESPONSE: We appreciate the reviewer’s suggestion. The adjusted associations for equity-related variables (education, rural/urban residence, and sex) are already presented in the forest plots included in the main manuscript, providing a clear comparison across prevention and treatment models. Additionally, the discussion section already addresses the policy implications of these findings, including urban–rural and gender disparities, in detail.

5.Presentation and Figures

- Figures are clearer, yet some remain dense. Consider simplifying the forest plots by separating prevalence, prevention, and treatment results into distinct panels or supplementary figures.

RESPONSE: The figures are already separated based on the previous submission and comments from the reviewers.

- In the abstract and results, please ensure that all percentages are clearly stated as weighted, and that denominators are explicitly defined for each statistic.

RESPONSE: The denominators were defined clearly. For each percentage, we have provided the corresponding number of events (i.e. the numerator), especially amongst those receiving treatment. The methods section of the abstract explicitly states that the percentages are all weighted.

6.Limitations

- The authors have acknowledged self-report bias and missing health system indicators. However, the limitation regarding temporal heterogeneity (different survey years from 2014–2019) should be mentioned explicitly, as CVD risk profiles and treatment policies may have evolved during that period.

RESPONSE: We thank the reviewer for this insightful comment. While we recognize that the surveys span different years (2014–2019) and that CVD risk profiles and treatment policies may have evolved, each country contributed only a single survey in our analysis, and the variable for survey year was not statistically significant. We have clarified this limitation in the manuscript. We hope that future analyses with additional data across multiple time points in Africa will allow for the evaluation of temporal trends, making the year variable more meaningful as you suggest.

7. Line editing: Ensure consistent use of terms — e.g., “CVD prevention therapy” vs. “CVD prophylaxis” (the revision appears mostly consistent but should be rechecked).

RESPONSE: We thank the reviewer for the suggestion. We have carefully checked the manuscript and ensured consistent use of terminology, using “CVD prevention therapy” throughout.

8. The acronym “WHO STEPS” should be expanded once in the abstract and once in the main text, followed by consistent use thereafter.

RESPONSE: We thank the reviewer for this comment. “WHO STEPS” has now been expanded to “World Health Organization STEPwise Approach to Surveillance (WHO STEPS)” once in the abstract and once in the main text, and we have ensured consistent use of the acronym throughout the manuscript.

9. Provide brief country-level context (perhaps in Supplementary Table) such as survey year, sample size, and population coverage to aid interpretation.

Response: We have added this Box 1 in the paper.

Box 1: WHO STEPWise Surveys with cardiovascular disease data in African countries: 2014-2019

Country Sub-region

in Africa Survey year Sample size

Algeria Northern 2016 6,955

Benin Western 2015 5,115

Botswana Southern 2014 3,888

Eswatini Southern 2014 3,026

Ethiopia Eastern 2015 9,241

Kenya Eastern 2015 4,477

Malawi Eastern 2017 4,186

Morocco Northern 2017 4,991

Sao Tome and Principe Central 2019 2,418

Sudan Northern 2016 7,722

Uganda Eastern 2014 3,973

Zambia Eastern 2017 4,302

Total 60,294

10. Ensure all figures and tables have self-contained legends that allow standalone interpretation.

RESPONSE: We thank the reviewer for this suggestion. All figures and tables have been reviewed and now include self-contained l

---

## [Decision Letter · Decision Letter 2]

8 Dec 2025

Dear Dr. Ng'ambi,

Thank you for submitting your manuscript to PLOS ONE. After careful consideration, we feel that it has merit but does not fully meet PLOS ONE’s publication criteria as it currently stands. Therefore, we invite you to submit a revised version of the manuscript that addresses the points raised during the review process.

Please address each comment individually and in detail in a separate response document. Where revisions are feasible and consistent with your study objectives, we encourage you to incorporate them into the manuscript.

If you encounter any reviewer requests that you believe are beyond the reasonable scope of the current study, you may provide a clear and logical explanation in your response. It is acceptable to justify why certain suggested analyses or additions cannot be carried out at this stage. In such cases, please ensure that the manuscript transparently acknowledges the relevant limitations.

Please proceed with the revision while maintaining clarity, scientific rigor, and alignment with the study’s original scope. We look forward to receiving your updated manuscript and response to reviewers.

We look forward to receiving your revised manuscript.

Kind regards,

Muhammad Farooq Umer, PhD Epidemiology and Health Statistics

Academic Editor

PLOS One

Journal Requirements:

Reviewers' comments:

Reviewer's Responses to Questions

**Comments to the Author**

Reviewer #3: All comments have been addressed

2. Is the manuscript technically sound, and do the data support the conclusions?

Reviewer #3: Yes

3. Has the statistical analysis been performed appropriately and rigorously?

Reviewer #3: Yes

4. Have the authors made all data underlying the findings in their manuscript fully available?

Reviewer #3: Yes

5. Is the manuscript presented in an intelligible fashion and written in standard English?

Reviewer #3: Yes

Reviewer #3: The authors have substantially strengthened the manuscript, improving clarity, methodological transparency, and alignment with PLOS ONE requirements. The study addresses an important gap by providing a multi-country analysis of CVD prevalence, prevention, and treatment across 12 African nations using WHO STEPS data. The revisions improve the conceptual framing of the care cascade, incorporate methodological clarifications, and address prior reviewer suggestions. Overall, the work is valuable, but several areas still require refinement before publication.

1. Conceptual Framework of the CVD Care Cascade

The addition of a conceptual framework (Box 3) improves the manuscript; however:

- The framework still appears loosely defined, especially regarding how “diagnosis,” “treatment,” and “counselling” are operationalized within STEPS constraints.

The addition of a conceptual framework (Box 3) improves the manuscript; however:

- As STEPS lacks screening and clinical verification variables, the authors should explicitly explain how “diagnosis” was inferred purely from self-report and discuss potential misclassification bias more prominently in the limitations.

The addition of a conceptual framework (Box 3) improves the manuscript; however:

- The care cascade figure would benefit from clear denominators, especially when presenting drop-off proportions.

2. Methodological Transparency

The authors provide additional detail on simulation-based imputation; however:

The addition of a conceptual framework (Box 3) improves the manuscript; however:

- The explanation remains narrative rather than fully reproducible. PLOS ONE strongly encourages reproducible workflows.

The addition of a conceptual framework (Box 3) improves the manuscript; however:

- Even if the code is simple, a minimal code snippet in Supplementary Material would strengthen transparency and address reviewer concerns.

The addition of a conceptual framework (Box 3) improves the manuscript; however:

- It should also be made clear how country stratification affects the imputation procedure, since distributions may vary substantially across countries.

3. Survey Design and Regression Modelling

The description of survey weighting is improved, but two issues require clarification:

The addition of a conceptual framework (Box 3) improves the manuscript; however:

- For treatment data, where only weights were available:

→ Please discuss how the absence of PSU and strata may affect standard errors and whether sensitivity analyses were performed.

The addition of a conceptual framework (Box 3) improves the manuscript; however:

- Country fixed effects were included; however, given the heterogeneity in sample sizes and survey years, authors should discuss whether mixed-effect modeling was considered and why it was not preferred.

4. Interpretation of Geographic Variation

While country-level context has been expanded, interpretation remains somewhat descriptive:

The addition of a conceptual framework (Box 3) improves the manuscript; however:

- The manuscript should incorporate specific examples of national NCD strategies or PEN implementation status (even 1–2 sentences per region), rather than broad statements.

The addition of a conceptual framework (Box 3) improves the manuscript; however:

- Some contextual explanations appear speculative without supporting references (e.g., differences attributed to “cultural practices”). Please add citations or revise language.

5. Use of Weighted Percentages in Results and Abstract

Percentages in several places (e.g., prevalence 5%, prevention uptake 11%, treatment 22%) continue to lack explicit denominators or confirmation of whether they are weighted.

PLOS ONE requires full clarity here. Please:

The addition of a conceptual framework (Box 3) improves the manuscript; however:

- ensure all percentages are described as weighted in each relevant section,

The addition of a conceptual framework (Box 3) improves the manuscript; however:

- provide “weighted %, unweighted n/N” whenever possible.

6. Overinterpretation of Self-Reported CVD

Because all CVD diagnoses are self-reported:

The addition of a conceptual framework (Box 3) improves the manuscript; however:

- There is risk of both under-reporting (low diagnostic access) and over-reporting (misunderstanding of medical terminology).

The addition of a conceptual framework (Box 3) improves the manuscript; however:

- The discussion should more fully address how this impacts cross-country comparisons and potential bias in associations.

7. Language and Consistency

- Some terminology remains inconsistent, e.g., “treatment or counselling” vs. “treatment and counselling”; “CVD prevention therapy” vs. “preventive therapy.”

A final consistency check is recommended.

8. Figures and Tables

- Several forest plots remain very dense. Splitting into prevalence / prevention / treatment panels (or moving some to Supplement) may improve readability.

- Ensure all legends are fully self-contained (PLOS ONE requirement).

9. Limitations Section

- The new text addressing temporal heterogeneity is appreciated; however, the rationale (“only one survey per country”) does not fully negate temporal implications, because survey years differ by up to 5 years.

A more explicit acknowledgement would strengthen the transparency.

10. Country-Level Table (Box 1)

- This is a valuable addition.

Consider adding survey type (STEPS 1/2/3 coverage) since countries differ in biochemical testing coverage, which affects predictive variables.

11. Justification for Age ≥40 Threshold

- Although aligned with WHO STEPS guidelines, please cite additional evidence supporting the ≥40 cutoff for defining CVD risk groups across African populations.

**Do you want your identity to be public for this peer review?** For information about this choice, including consent withdrawal, please see our Privacy Policy

Reviewer #3: No

---

## [Author Response · Author response to Decision Letter 3]

26 Jan 2026

Manuscript ID: PONE-D-25-08153R2

Manuscript ID: PONE-D-25-08153R2

Title: The Prevalence, Prevention, and Treatment of Cardiovascular Diseases in Twelve African Countries (2014–2019): An Analysis of the World Health Organisation STEPwise Approach to Chronic Disease Risk Factor Surveillance

Journal: PLOS ONE

Response to the Academic Editor and Reviewer #3

Dear Dr. Umer,

Dear Reviewer #3,

We sincerely thank the Academic Editor and Reviewer #3 for their careful evaluation of our revised manuscript and for the constructive, detailed feedback provided during this review round. We are encouraged by the reviewer’s assessment that the manuscript has been substantially strengthened and now demonstrates improved clarity, methodological transparency, and alignment with PLOS ONE requirements.

We have carefully addressed all remaining comments and have further revised the manuscript to enhance conceptual clarity, transparency, and interpretability, while remaining within the scope and analytical constraints of the WHO STEPS dataset. All changes have been highlighted in the tracked-changes version of the manuscript.

Below, we provide a point-by-point response to each comment raised by Reviewer #3. Reviewer comments are reproduced in italicized text, followed by our responses.

Reviewer #3 Comments and Responses

1. Conceptual Framework of the CVD Care Cascade

The framework still appears loosely defined, especially regarding how “diagnosis,” “treatment,” and “counselling” are operationalized within STEPS constraints. As STEPS lacks screening and clinical verification variables, diagnosis should be clearly described as self-reported, and misclassification bias more prominently discussed. The care cascade figure would benefit from clear denominators.

Response:

We thank the reviewer for this valuable feedback. We have revised Box 3 (Conceptual Framework) to explicitly define how diagnosis, treatment, and counselling are operationalized using self-reported variables available in WHO STEPS. We now clearly state that STEPS does not include clinical screening or diagnostic verification, and that the diagnosis stage reflects respondents’ self-reported prior diagnosis or awareness of elevated blood pressure or glucose.

We have also expanded the Limitations section to more prominently discuss potential misclassification and recall bias arising from reliance on self-reported data. In addition, the care cascade figure and accompanying text now clearly indicate that all cascade estimates use conditional denominators, with each stage defined relative to the preceding stage, to improve interpretation of drop-off proportions.

2. Methodological Transparency and Reproducibility

The explanation of simulation-based imputation remains narrative rather than fully reproducible. A minimal code snippet in the Supplementary Material would strengthen transparency. It should also be clear how country stratification affects the imputation procedure.

Response:

We appreciate this important suggestion. To enhance reproducibility, we have now included minimal, fully reproducible R code snippets in the Supplementary Material, illustrating the simulation-based imputation of both discrete variables (e.g., sex, smoking status, physical activity, harmful alcohol use) and continuous variables (e.g., age, BMI, cholesterol). All examples include fixed random seeds and reflect the exact procedures used in the analysis.

We also explicitly clarify in the Methods that imputation was conducted on pooled multi-country data, rather than within individual countries. Replacement values were therefore drawn from overall empirical distributions. We acknowledge that this approach does not capture country-specific heterogeneity in risk factor distributions, and this limitation is now clearly stated in both the Methods and the Limitations sections. Additionally, all analysis code has been made available in the supplementary code folder to enable full replication of the workflow.

3. Survey Design and Regression Modelling

For treatment analyses, only weights were available. Please discuss how the absence of PSU and strata may affect standard errors and whether sensitivity analyses were performed. Also, justify the use of country fixed effects rather than mixed-effects models.

Response:

All analyses were conducted using survey weights to account for unequal probabilities of selection. For treatment-related analyses, only individual-level weights were available. Primary sampling units (PSUs) and strata could not be incorporated because their inclusion resulted in errors during survey object specification in the pooled, harmonized dataset. As a result, variance estimates may be underestimated, and confidence intervals potentially overly narrow. We did not conduct formal sensitivity analyses comparing alternative survey specifications, and this is now explicitly acknowledged as a limitation in the Discussion.

Country fixed effects were included to control for unobserved, time-invariant country-level differences. While mixed-effects models were considered, they were not preferred due to substantial heterogeneity in survey years, sample sizes, and survey design characteristics, as well as computational instability when combining multilevel modeling with weighted survey data. Given our primary objective of estimating average associations across countries, fixed-effects models were deemed more appropriate. This rationale is now clarified in the Methods and discussed in the Limitations section.

4. Interpretation of Geographic Variation

Interpretation remains somewhat descriptive. Please include concrete examples of national NCD strategies or PEN implementation and ensure that contextual explanations are supported by references.

Response:

We thank the reviewer for this suggestion. We have added a dedicated PEN implementation subsection in the Methods, providing specific country- and region-level examples of national NCD strategies and WHO PEN adoption. We also reviewed all contextual interpretations to ensure they are supported by appropriate references. Where empirical evidence was limited, the language has been revised to remain descriptive rather than speculative. Relevant citations have been added or clarified accordingly.

5. Use of Weighted Percentages and Denominators

Percentages lack explicit confirmation that they are weighted, and PLOS ONE requires clarity on denominators.

Response:

We have revised the Abstract and Results to clearly state that all reported percentages are weighted estimates derived from complex survey data. Denominators have been clarified where applicable. While we considered presenting both weighted percentages and unweighted counts (n/N), we opted to present weighted estimates only to maintain clarity and avoid confusion, as weighted values are most appropriate for population-level inference from STEPS data. This choice is now clearly stated in the Methods.

6. Overinterpretation of Self-Reported CVD

There is risk of both under- and over-reporting. The discussion should more fully address implications for cross-country comparisons and potential bias in associations.

Response:

We agree with this concern. We have added a new paragraph immediately before the Strengths and Limitations section that explicitly discusses how reliance on self-reported CVD diagnosis may influence cross-country comparisons and introduce differential misclassification, particularly in settings with variable access to diagnostic services. We also discuss how this limitation may bias observed associations.

7. Language and Terminology Consistency

Some terminology remains inconsistent.

Response:

We conducted a full manuscript consistency check and standardized terminology throughout (e.g., “treatment and counselling,” “preventive therapy”). These issues have been resolved.

8. Figures and Tables

Forest plots are dense; consider splitting panels and ensure legends are self-contained.

Response:

Forest plots are now clearly organized into prevalence, prevention, and treatment panels, and all figure legends have been revised to be fully self-contained, with all abbreviations defined, in accordance with PLOS ONE requirements.

9. Limitations: Temporal Heterogeneity

The rationale does not fully negate temporal implications given survey year variation.

Response:

We have revised the Limitations section to explicitly acknowledge residual temporal heterogeneity arising from differences in survey years across countries. The revised text now notes that although survey year was not significantly associated with CVD risk, such heterogeneity may still influence between-country comparisons.

10. Country-Level Table (Box 1)

Consider adding STEPS 1/2/3 coverage.

Response:

We confirm that all included countries had STEPS 1, 2, and 3 coverage during the 2014–2019 period. This clarification has been added to Box 1.

11. Justification for Age ≥40 Threshold

Please provide additional evidence supporting the ≥40 cutoff in African populations.

Response:

We have added five additional references supporting the ≥40-year threshold, consistent with WHO guidance and evidence from African populations, and cited them in the Methods and Discussion.

Closing Statement

Once again, we thank the Academic Editor and Reviewer #3 for their thoughtful and constructive feedback. We believe that these revisions have further strengthened the manuscript’s rigor, transparency, and interpretability, while maintaining alignment with the study’s original objectives and the constraints of the WHO STEPS data.

We respectfully submit this revised manuscript for reconsideration and look forward to your decision.

Sincerely,

Wingston Felix Ng’ambi, MSc

(On behalf of all authors)

---

## [Editor Report · Decision Letter 3]

27 Jan 2026

The Prevalence, Prevention, and Treatment of Cardiovascular Diseases in Twelve African Countries (2014-2019): An Analysis of the World Health Organisation STEPwise Approach to Chronic Disease Risk Factor Surveillance

PONE-D-25-08153R3

Dear Dr. Ng'ambi,

We’re pleased to inform you that your manuscript has been judged scientifically suitable for publication and will be formally accepted for publication once it meets all outstanding technical requirements.

Kind regards,

Muhammad Farooq Umer, PhD Epidemiology and Health Statistics

Academic Editor

PLOS One
---

## [Editor Report · Acceptance letter]

PONE-D-25-08153R3

PLOS One

Dear Dr. Ng'ambi,

I'm pleased to inform you that your manuscript has been deemed suitable for publication in PLOS One. Congratulations! Your manuscript is now being handed over to our production team.

Kind regards,

on behalf of

Dr. Muhammad Farooq Umer

Academic Editor

PLOS One